# Object vision to hand action in macaque parietal, premotor, and motor cortices

Stefan Schaffelhofer[1,2], Hansjörg Scherberger[1,3*]

[1]Neurobiology Laboratory, German Primate Center GmbH, Göttingen, Germany; [2]Laboratory of Neural Systems, The Rockefeller University, New York, United States; [3]Department of Biology, University of Göttingen, Göttingen, Germany

**Abstract** Grasping requires translating object geometries into appropriate hand shapes. How the brain computes these transformations is currently unclear. We investigated three key areas of the macaque cortical grasping circuit with microelectrode arrays and found cooperative but anatomically separated visual and motor processes. The parietal area AIP operated primarily in a visual mode. Its neuronal population revealed a specialization for shape processing, even for abstract geometries, and processed object features ultimately important for grasping. Premotor area F5 acted as a hub that shared the visual coding of AIP only temporarily and switched to highly dominant motor signals towards movement planning and execution. We visualize these non-discrete premotor signals that drive the primary motor cortex M1 to reflect the movement of the grasping hand. Our results reveal visual and motor features encoded in the grasping circuit and their communication to achieve transformation for grasping.

## Introduction

Grasping objects of different shapes and sizes appears trivial in daily life. We can distinguish between thousands of objects (*Biederman, 1987*) and shape our hands according to their geometry in order to hold and manipulate them (*Napier, 1956*; *Smeets and Brenner, 1999*). Although such operations seem to be effortless, their underlying neuronal mechanisms are highly complex and require extensive computational resources (*Fagg and Arbib, 1998*; *Felleman and Van Essen, 1991*). The cortical grasping network needs to translate high-dimensional visual information of an object into multidimensional motor signals that control the complex biomechanics of the hand.

In the primate brain, these processes are linked to the anterior intraparietal (AIP), the ventral premotor (F5), and the primary motor cortex (M1) (*Brochier and Umilta, 2007*; *Castiello, 2005*; *Davare et al., 2011*; *Nelissen and Vanduffel, 2011*). Within this network, AIP provides access to the dorsal visual stream that processes vision for action (*Culham et al., 2003*; *Goodale et al., 1994*). In fact, neurons in AIP were shown to strongly respond to the presentation of graspable objects or 3D contours (*Murata et al., 2000*; *Taira et al., 1990*; *Theys et al., 2012b*), but could also encode specific grip types (*Baumann et al., 2009*). This grasp-relevant information processed in AIP is exchanged with F5 via dense reciprocal connections (*Borra et al., 2008*; *Gerbella et al., 2011*; *Luppino et al., 1999*). Accordingly, deactivation of AIP or F5 causes severe deficits in pre-shaping the hand while approaching an object (*Fogassi et al., 2001*; *Gallese et al., 1994*). In contrast to AIP, concurrent electrophysiological studies suggest that F5 is primarily encoding objects in motor terms and is storing context-specific grip type information (*Fluet et al., 2010*; *Raos et al., 2006*). Connections of the dorsal subdivision of F5 (F5p) to the spinal cord and to M1 provide further evidence for the important role of F5 for grasp movement preparation (*Borra et al., 2010*; *Dum and Strick, 2005*).

*For correspondence: hscherb@gwdg.de

Competing interests: The authors declare that no competing interests exist.

**eLife digest** In order to grasp and manipulate objects, our brains have to transform information about an object (such as its size, shape and position) into commands about movement that are sent to our hands. Previous work suggests that in primates (including humans and monkeys), this transformation is coordinated in three key brain areas: the parietal cortex, the premotor cortex and the motor cortex. But exactly how these transformations are computed is still not clear.

Schaffelhofer and Scherberger attempted to find out how this transformation happens by recording the electrical activity from different brain areas as monkeys reached out to grasp different objects. The specific brain areas studied were the anterior intraparietal (AIP) area of the parietal cortex, a part of the premotor cortex known as F5, and the region of the motor cortex that controls hand movements. The exact movement made by the monkeys' hands was also recorded.

Analysing the recorded brain activity revealed that the three brain regions worked together to transform information about an object into commands for the hand, although each region also had its own specific, separate role in this process. Neurons in the AIP area of the parietal cortex mostly dealt with visual information about the object. These neurons specialized in processing information about the shape of an object, including information that was ultimately important for grasping it. In contrast, the premotor area F5 represented visual information about the object only briefly, quickly switching to representing information about the upcoming movement as it was planned and carried out. Finally, the neurons in the primary motor cortex were only active during the actual hand movement, and their activity strongly reflected the action of hand as it grasped the object.

Overall, the results presented by Schaffelhofer and Scherberger suggest that grasping movements are generated from visual information about the object via AIP and F5 neurons communicating with each other. The strong links between the premotor and motor cortex also suggest that a common network related to movement executes and refines the prepared plan of movement. Further investigations are now needed to reveal how such networks process the information they receive.

These electrophysiological and anatomical observations lead to our current understanding of the fronto-parietal network as the main circuitry for translating object attributes into motor commands (*Jeannerod et al., 1995*; *Rizzolatti and Luppino, 2001*). In detail, it has been suggested that visual features extracted in AIP activate motor prototypes in F5, which store hand configurations according to an object's geometry (*Rizzolatti and Luppino, 2001*). However, the detailed neural mechanisms of these processes remained unclear.

To create a deeper understanding of how visual information is transformed into motor commands, a precise identification and differentiation of visual and motor processes within the grasping network is required. Previous important grasping studies classified visual-dominant, visual-motor or motor-dominant neurons primarily on the phase of their activation [for AIP see *Murata et al. (2000)* and *Sakata et al. (1995)*, for F5 see *Raos et al. (2006)*; *Theys et al. (2012a)*], but they could not discriminate between neural coding of visual features of objects or motor features of the hand.

A differentiation between visual and motor coding is challenging for multiple reasons. First, the fronto-parietal network is multimodal and can reflect sensory and motor signals simultaneously. Second, visual and motor descriptions of objects and the hand are multidimensional due to the complexity of object geometry and hand physiology. Investigations at the neural level therefore necessitate multidimensional observations from many neurons. Third, the visual and motor spaces are highly linked to each other since the form of an object often defines the shape of the grasping hand. Disassociating both neuronal representations therefore requires highly variable visual stimuli and motor responses.

In this study, we took a multidimensional approach to identify and separate visual and motor processes in the grasping network of AIP, F5, and M1. We recorded large populations of neurons simultaneously from the entire network and compared their modulation patters to the visual attributes of highly diverse objects and the kinematic features recorded from the grasping hand. Our data revealed distinct roles of the grasping network in translating visual object attributes (AIP) into

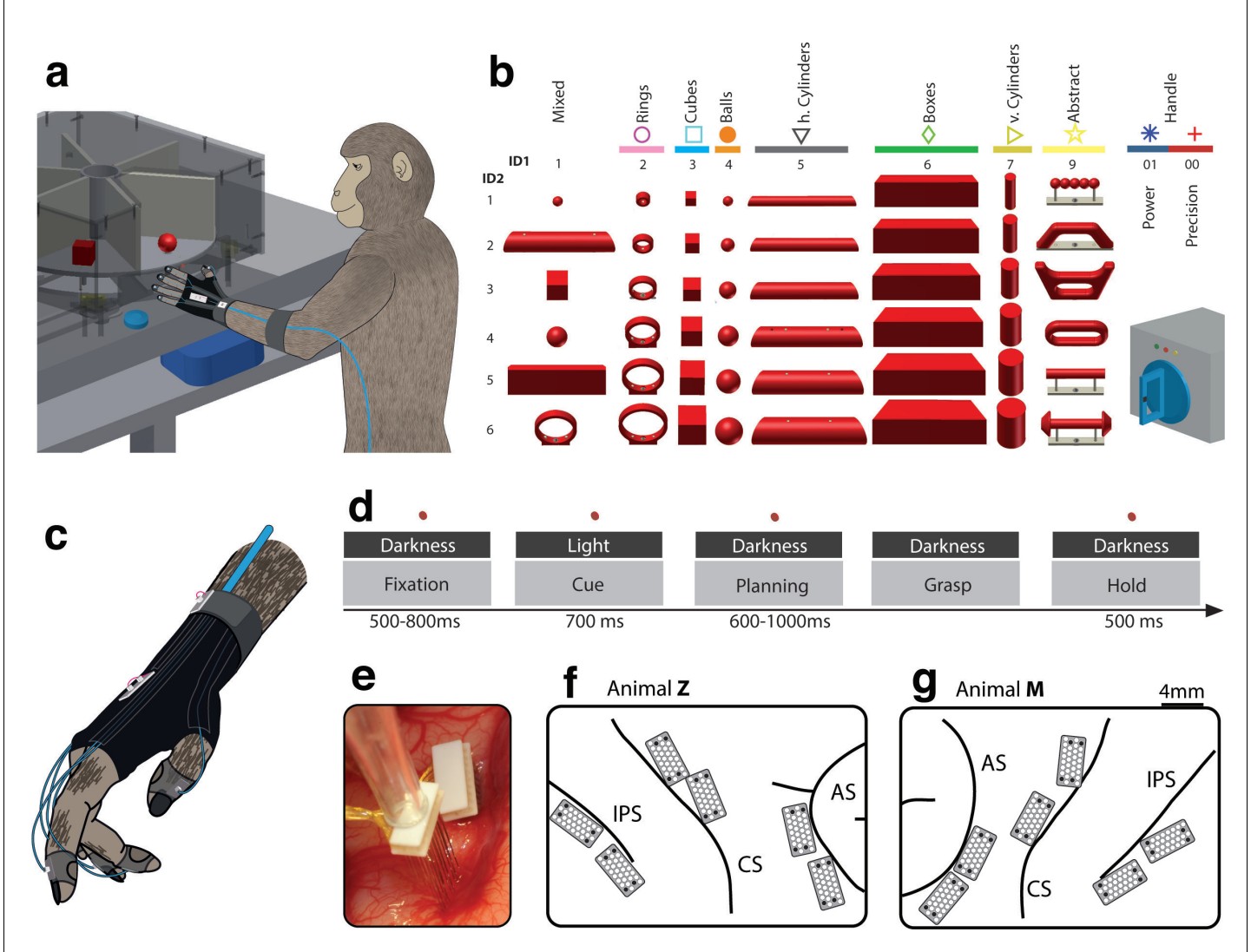

**Figure 1.** Behavioural design and implantation details. (a–b) Two monkeys were trained to grasp a total of 48 objects presented on a PC-controlled turntable. In addition, monkeys were instructed to perform either precision or power grips on a handle. Each of the 50 grasping conditions was denoted with a double-digit number (ID1, ID2), a colour code, and a symbol to allow easy identification throughout this manuscript. (c) An instrumented glove equipped with electro-magnetic sensors allowed monitoring and recording the monkeys' hand and arm kinematics in 27 DOF. (d) All grasping actions were performed as a delayed grasp-and-hold task consisting of eye-fixation, cue, planning, grasping and hold epochs. (e–g) Neural activity was recorded simultaneously from six floating microelectrode arrays implanted in the cortical areas AIP, F5, and M1. (f) Electrode placement in monkey Z (right hemisphere). Each array consisted of 2 ground and 2 reference electrodes (black), as well as 32 recording channels (white) aligned in a 4x9 matrix. Electrode length for each row increased towards the sulcus from 1.5–7.1 mm. (g) Same for monkey M (left hemisphere). Two arrays were implanted in each area. AIP: toward the lateral end of the intraparietal sulcus (IPS); F5: on the posterior bank of the arcuate sulcus (AS); hand area of M1: on the anterior bank of the central sulcus (CS).

planning (F5) and execution signals (M1) and allowed visualizing the propagation of these features for grasping.

## Results

Two macaque monkeys grasped a large set of 49 objects causing highly variable visual stimuli and motor responses (*Figure 1a–b*, *Video 1*). During the experiments we recorded hand and arm kinematics from an instrumented glove (*Schaffelhofer and Scherberger, 2012*) (see *Figure 1c*, *Video 2*)

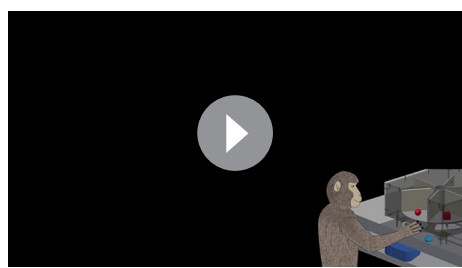

**Video 1.** Experimental task. A monkey grasped and held highly variable objects presented on a PC-controlled turntable. Note: For presentation purposes, the video was captured in the light.

in conjunction with neuronal activity from 6 cortical microelectrode arrays (6 x 32 channels) (*Figure 1e–g*).

Training the monkeys to perform grasping movements in a delayed grasp-to-hold paradigm allowed us to investigate neural activity at several key stages of the task. As shown in *Figure 1d*, visual responses (i.e., object presentation in cue epoch), planning activity (i.e., motor preparation in planning epoch), and motor execution signals (i.e., grasp and hold epoch) were temporarily distinct and could therefore be explored separately.

We analysed data from 20 recording sessions of two macaque monkeys (10 sessions per animal). On average, spiking activity of 202 ± 7 and 355 ± 20 (mean ± s.d.) single and multiunits were collected in each session in monkey Z and M, respectively. Of these units, 29.2% and 25.2% were recorded from AIP, 37.3% and 32.3% from F5, and 33.5% and 42.5% from M1 (monkey Z and M, respectively).

## Vision for hand action

Presenting 3D objects to the monkeys lead to vigorous discharge (*Baumann et al., 2009*; *Murata et al., 2000*) of AIP-neurons (*Figure 2a–b*). The modulated population was not only larger (sliding ANOVA, *Figure 3*), but also significantly faster appearing after stimulus onset than in F5 (49.7 ms and 54.9 ms, monkey M and Z respectively). Impressively, individual AIP cells were capable of differentiating object shapes at high precision (e.g., *Figure 2a*). To quantify this attribute, we computed a modulation depth analysis that determined the relative difference in firing rate between all pairs of conditions (objects) during the cue epoch (see Materials and methods). The example cell of *Figure 2b* revealed a chequered structure caused by the shape-wise order of object conditions 00–76 (for object id, see *Figure 1b*) and a maximum modulation depth (MD) of 62 Hz. Statistical analysis between all conditions (ANOVA and post-hoc Tukey-Kramer criterion, p<0.01; see Materials and methods) revealed a high encoding capacity of the example neuron that could significantly separate 71% of the 946 condition pairs (44 conditions). Interestingly, the neuron decreased its MD in darkness but maintained its encoding of shape (as also indicated in *Figure 2a*).

To investigate this effect at the neuronal population level, we performed canonical discriminant analysis (CDA; see Materials and methods), which allowed reducing the neuronal state space (N-space) to its most informative dimensions. *Figure 2c and 2d* show the first three canonical variables during the cue and grasp epoch, respectively. In them, each marker represents the neuronal state of an individual trial in the AIP population (see *Figure 1b* for symbol and colour code). In this N-space of AIP, we found objects to be separated based on their shape. Independent of the way the objects were grasped, the neural space accurately differentiated horizontal cylinders (black), vertical cylinders (green), rings (magenta), spheres (orange), cubes (blue), and bars (black) (see *Video 3* for an animated 3D view of a typical N-space).

To further quantify these findings, we computed the Mahalanobis distance between all pairs of conditions in the complete N-space of AIP (see Materials and methods). Hierarchical cluster analysis (HCA) performed on these distance measures confirmed the findings of the CDA and revealed a clear clustering according to object shape during visual presentation of the object (*Figure 2e*)

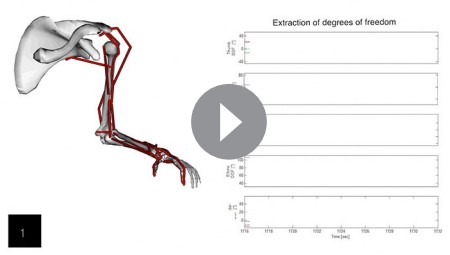

**Video 2.** Hand and arm tracking. 18 joints of the primate hand were tracked with electromagnetic sensors and used to drive a 3-D primate-specific musculoskeletal model to extract 27 joint angles. Thumb, index, wrist, elbow, and shoulder angles are shown while the monkey is grasping a ring, a ball and a cylinder. The video runs at half speed.

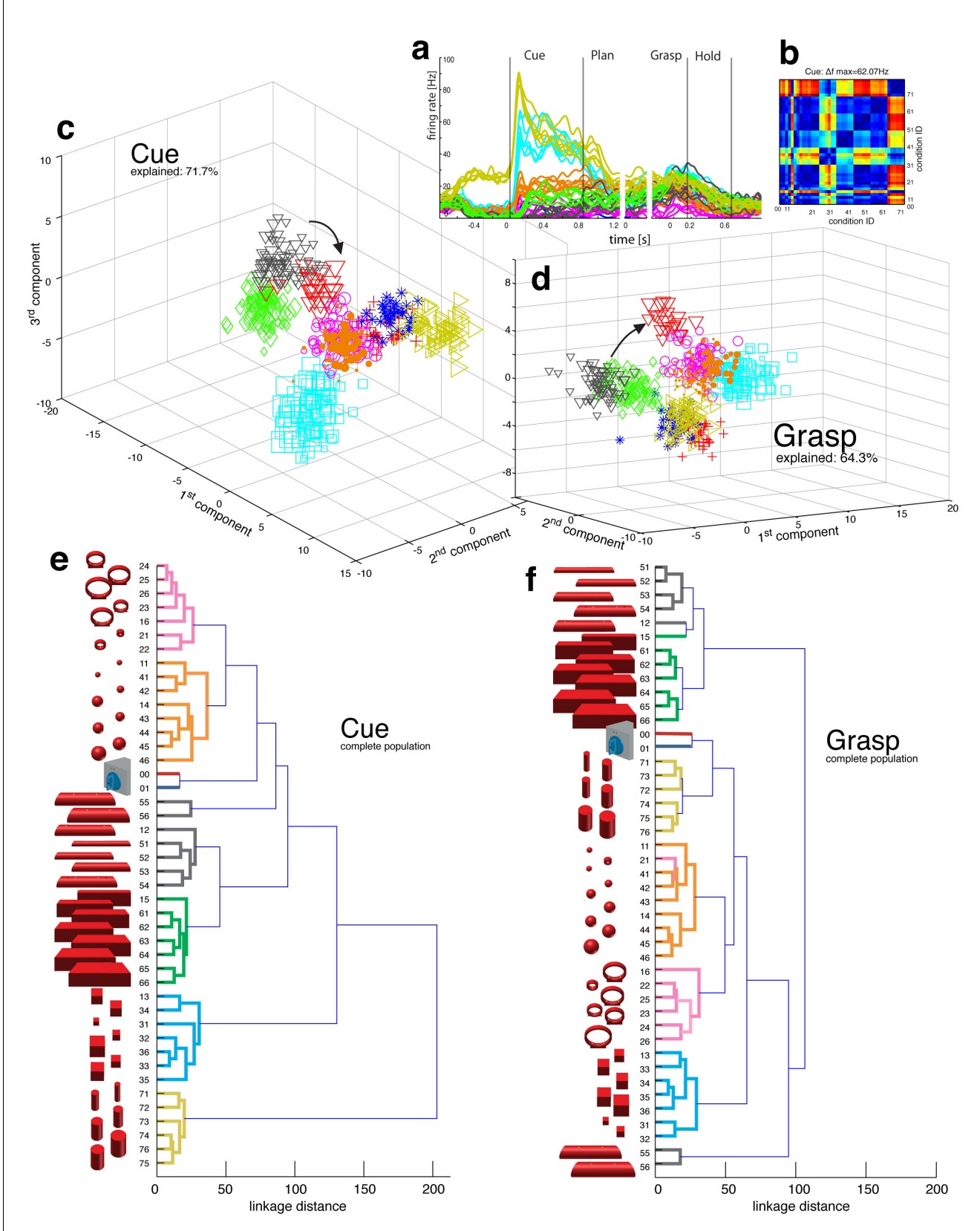

**Figure 2.** Visual object processing in area AIP. (**a**) Example neuron of AIP responding to the presentation of graspable objects (each curve represents one task condition). (**b**) Modulation depth plot illustrating the absolute firing rate difference in the cue epoch between all condition pairs (conditions 00

*Figure 2 continued on next page*

*Figure 2 continued*

– 76 placed on axis in ascending order). Warm colours: high modulation depth, cool colours: low modulation depth. (c) Shape-wise clustering of objects in the AIP population during the cue epoch, as demonstrated by CDA. Arrows indicate a shift in position when big horizontal cylinders (red triangles) were grasped from below instead from above (black triangles). (d) Same as c, but during the grasp epoch. (e–f) Dendrograms illustrating the neural distance between object conditions in the simultaneously recorded AIP population in the cue and grasp epoch (N = 62). Symbols and colour code in a, c-f as in *Figure 1b*. Percentages in c and d describe how much variance of the data is explained by the shown components (1st, 2nd and 3rd). Note: *Video 3* visualizes the N-space of AIP of an additional recording in the same animal (Z). See *Figure 2—figure supplement 1–2* for the averaged population results of animal Z and animal M, respectively.

The following figure supplements are available for figure 2:

**Figure supplement 1.** Visual coding for hand action in AIP in animal Z across all sessions.

**Figure supplement 2.** Visual coding for hand action in AIP in animal M across all sessions.

that widely persisted during movement execution, although with significantly shorter neural distances (*Figure 2f*). 100% and 91% of the objects shared their cluster with other objects of the same shape during the cue and grasp epoch, respectively. Importantly, consistent results were observed in both monkeys when performing HCA across all recording sessions (*Figure 2—figure supplement 1–2*, see Materials and methods).

The large number of objects presented in one recording session required separating the 48 objects on different turntables (see *Figure 1*), often objects of similar shape. This separation created small offsets already in the fixation epoch, but at very low modulations. An extreme case is shown in *Figure 2a*. This might raise concerns on whether coding in AIP was really due to object shape, or rather to the object presentation order (different turntables presented sequentially). However, shape-wise clustering in AIP cannot be explained by the task design for the following reasons: (1) The offsets in the fixation epoch were very small in comparison to the visual modulations observed in AIP when the objects were presented (e.g., see *Video 5*). (2) The set of 'mixed' objects – presented and grasped in the same block – clustered with other objects of the same shape (e.g. ring in mixed block clusters with other ring objects). Together, this demonstrates clear shape processing in AIP.

The AIP population also encoded the size of objects, but differentiated this geometrical feature at clearly lower neural modulations compared to object shape. As shown in *Figure 2e–f* and supplements, the majority of objects were located closest to objects of similar size, but with significantly shorter neural distances in comparison to shape features. The secondary role of size was surprising, since object size has significant influence on the aperture of the grasping hand (*Jakobson and Goodale, 1991*).

To further test the specialisation for shape processing, we tested a mixed set of objects (*Figure 3a*), causing highly variable visual stimuli and motor responses, against an abstract object set (*Figure 3d*) that we have specifically designed to cause different visual stimuli but the same grip. In theory, a purely visual area should differentiate both sets of shapes, since they both provide different visual stimuli. In contrast, an exclusive motor area should not show modulations for the abstract objects because they require the same grip. Strikingly, the AIP population responded highly similar to the presentation (cue epoch) of mixed (*Figure 3b*) and abstract (*Figure 3e*) objects (t-test, p>0.05; 35% vs. 35% and 20% vs. 17% comparing mixed and abstract objects in monkey Z and M, respectively). Thus, the equal responses to both object sets strongly

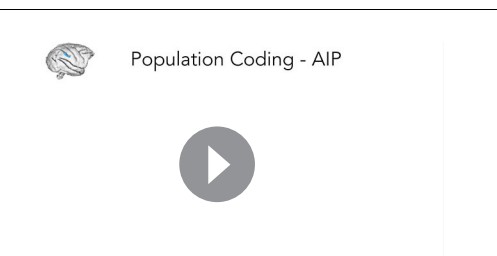

**Video 3.** Population coding in AIP. The first three canonical variables of the AIP population are shown in 3D and are animated for presentation purposes. Each symbol represents one trial. Symbols and colours as in *Figure 1b*.

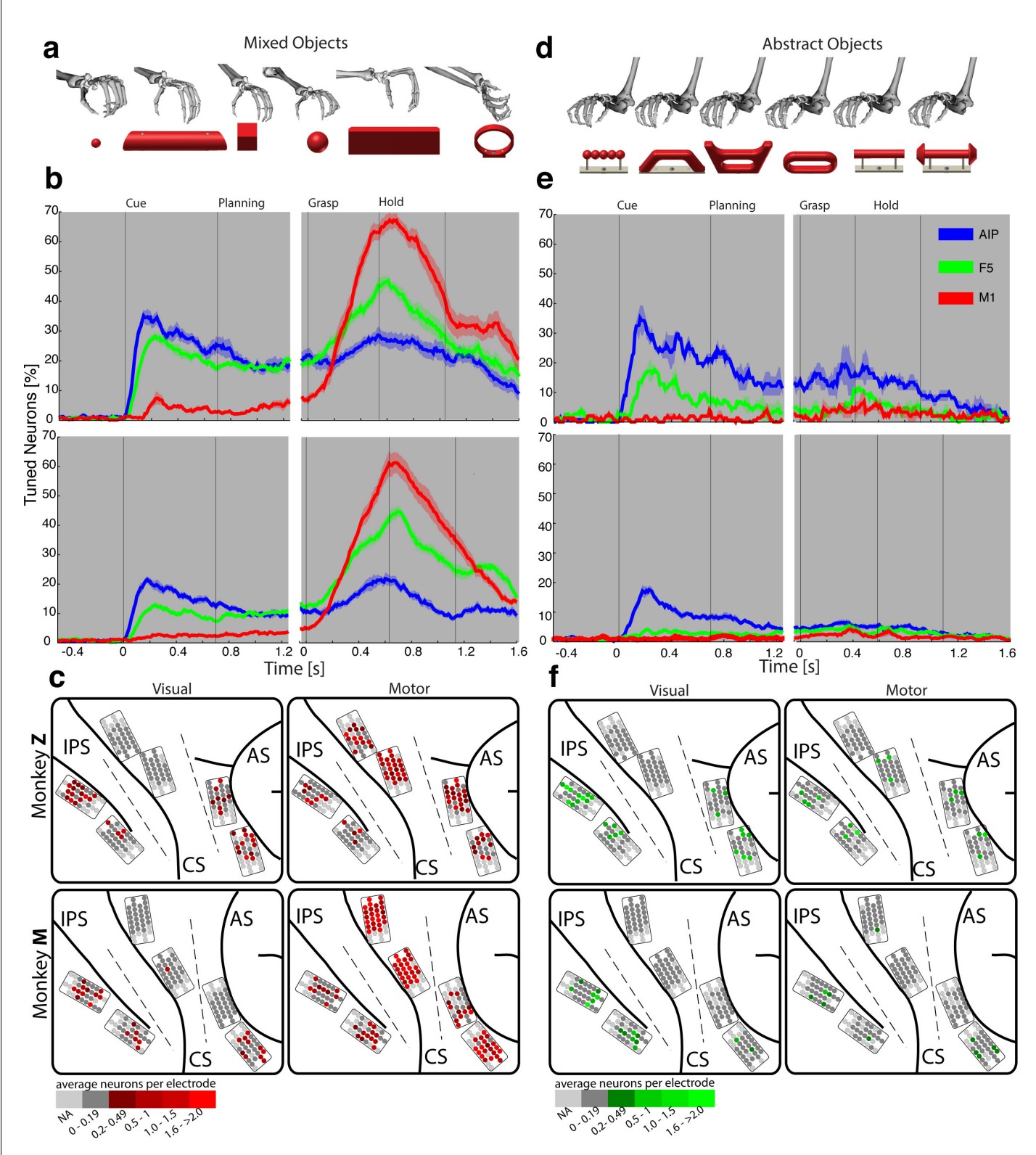

**Figure 3.** Visual processing of object shapes. (**a**) A set of six 'mixed' objects elicited different visual stimuli and different motor responses. (**b**) Percentage of tuned neurons of the AIP, F5, and M1 population express the significant modulation with respect to the mixed objects across time (sliding one-way ANOVA). (**c**) Tuned neurons (shades of red) were mapped to their recording location during the visual (t = 0.16 s after object presentation) and motor phase (t = 0.7 s after movement onset). (**d**) As a contrast and to elicit pure visual responses, 'abstract' objects caused different

*Figure 3 continued on next page*

*Figure 3 continued*

visual stimuli but the same grip. (**e**) Similar to b, but for the abstract objects set. (**f**) Similar to c, but showing the map of tuned neurons (shades of green) with respect to the abstract object set. For b, e: Data is doubly aligned on cue onset and on the grasp (go) signal. Sliding ANOVA was computed for each session individually and averaged across all 10 recording sessions per animal. Shades represent standard error from mean (s.e.m.) across recording sessions. For c, f: The number of tuned neurons per channel were averaged across all recording sessions and visualized in shades of green and red for the abstract and mixed objects, respectively. Channels without any identified neurons were highlighted in light grey. Map of monkey M is mirrored along vertical axis for better comparison of both animals.

supports the specialisation of AIP in processing object shapes.

Importantly, AIP remained the most tuned area when the monkeys planned and grasped the abstract objects, as shown in *Figure 3e*. However, the number of significantly tuned cells decreased during these epochs in comparison to the responses evoked by the mixed objects (*Figure 3b*). This reduced selectivity could indicate either motor or visual transformations that are both required for grasping: First, the abstract objects were grasped with the same hand configuration (see *Figure 3d*). Unmodulated activity could therefore reflect the same motor affordance across the six abstract objects (*Fagg and Arbib, 1998*; *Rizzolatti and Luppino, 2001*). Second, the abstract objects have different shapes, but have the graspable handle in common that shares the same geometrical dimensions across all six objects (see *Figure 3d*). A uniform modulation could therefore also represent visual processes that reduce the objects to its parts relevant for grasping (i.e. the same geometries of the handle across the six abstract objects). Considering that the same AIP population separated the complete object set primarily on their geometrical features (*Figure 2e–f*) would suggest visual rather than motor transformations explaining uniform modulations.

We found further indicators for this hypothesis when focusing on objects that caused, in contrast to the abstract objects, equal visual stimuli but different motor responses. To create such a scenario, monkeys were trained to perform power or precision grips on the same object, the handle (condition 00 and 01). Although both conditions were located most distantly in the kinematic or joint-angle (J-) space in both monkeys (see *Figure 4e* and *5a*), they were located closest to each other in the N--space of AIP (see *Figure 2e–f*), clearly implying a visual representation of the handle. Statistical analysis revealed, however, that both conditions (00, 01) slightly increased their neural distance (*Figure 2d*) towards planning and movement execution, as expressed by an average of 21% and 16% of significantly tuned AIP neurons in monkey Z and M (ANOVA tested in grasp epoch, p<0.01). These observations suggest a visual representation of the handle and a further differentiation of its parts that are relevant for grasping.

Similarly, the AIP population coded all horizontal cylinders based on their shape and then differentiated the two biggest horizontal cylinders (55,56, see *Figure 2*), when they were grasped differently (either from top or from below). Both conditions (55, 56) required a focus on different parts of the object (top vs. bottom edge) as well as different hand configurations (pronation vs. supination). Importantly, the neural representation of both conditions originated from the very same shape cluster in N-space (*Figure 2c,e*) that subsequently drifted further apart (see arrows in *Figure 2d,f*). Together, these observations suggest a visual rather than motor representation in AIP.

## Motor planning and execution

To generate grasping movements, visual attributes of objects need to be transformed into adequate motor commands before they get executed (*Jeannerod et al., 1995*; *Rizzolatti and*

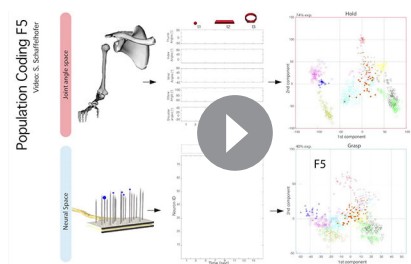

**Video 4.** Population coding in F5. Joint angles and the population activity of F5 were recorded together and for visual display reduced to their most informative dimensions (component 1 and 2). The video displays the evolving hand kinematics (top, left) and neuronal population activity (bottom, left) during three subsequent grasping actions. Arrows point at these trials in the J- (top, right) and N-space (bottom, right). The audio-track plays the spiking activity of an individual F5 neuron, which is highlighted in the raster plot in blue.

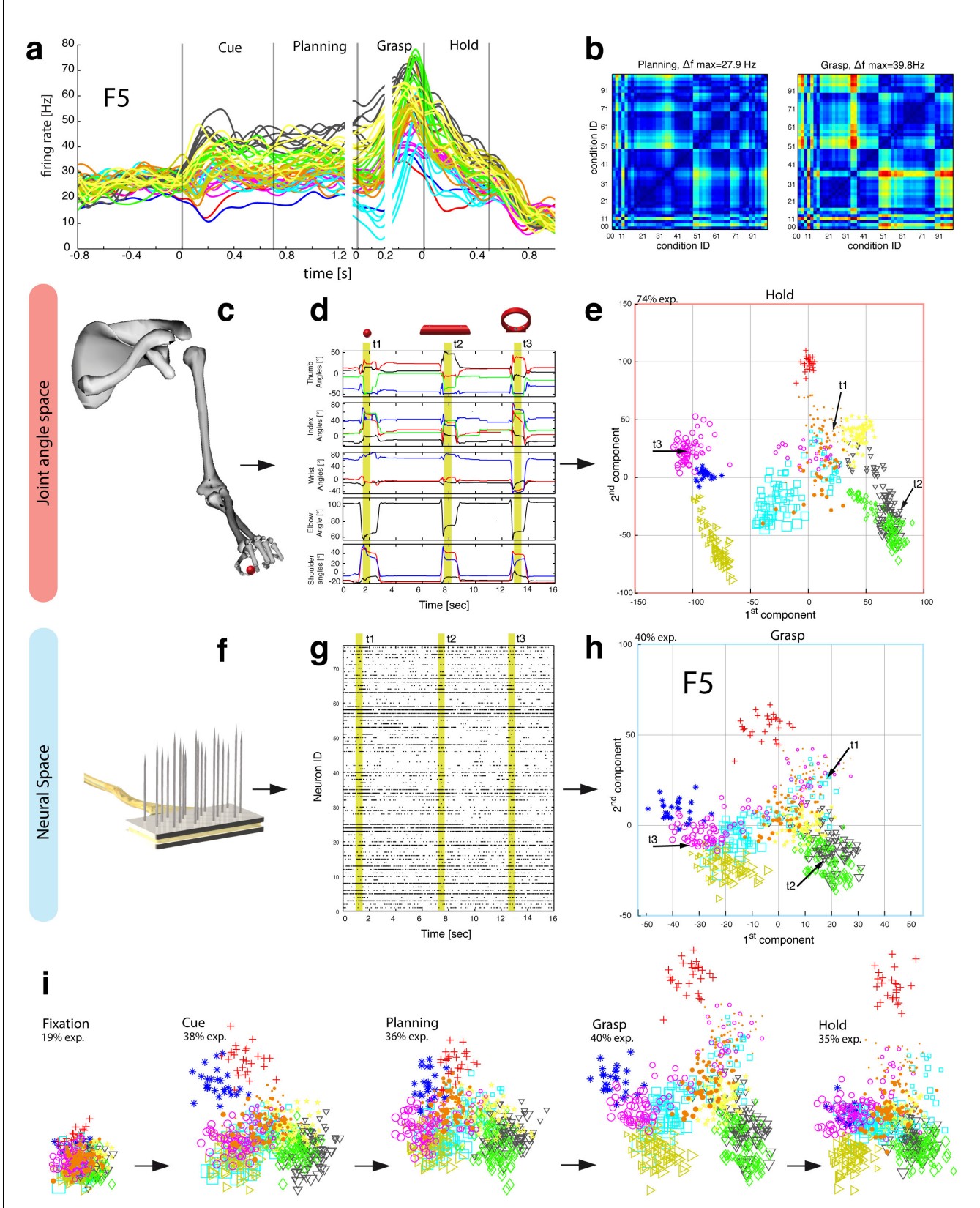

**Figure 4.** Motor planning and execution in F5. (a) Example neuron of F5, responding to all 50 task conditions (colour code as in *Figure 1b*). (b) Modulation depth plots (as in *Figure 2b*) in the planning and grasp epoch. (c) Recorded kinematics was used to drive a monkey-specific

*Figure 4 continued on next page*

*Figure 4 continued*

musculoskeletal model that allowed extracting 27 DOF. (**d**) A selection of DOFs is presented for three sequential grips: thumb and index finger, wrist, elbow, and shoulder. (**e**) PCA performed on the J-space during the hold epoch allowed visualizing the grip types of all conditions and trials of one recording session (showing 1$^{st}$ and 2$^{nd}$ PCA component). (**f–g**) Raster plot shows the spiking activity of F5-neurons recorded from a single FMA (F5-ventral). (**h**) Mean firing rates during the grasp epoch (N-space) were transformed with CDA to reduce and visualize the multidimensional representation of the complete F5 population (N = 119, simultaneously recorded). In d,g, example trials t1, t2 and t3 are highlighted in yellow (hold epoch in d, grasp epoch in g) and marked with arrows in e,h. (**i**) Neuronal state space evolution shows the course of the task determined by the CDA. e,h and i: For visual comparison the N-space was aligned to the J-space using PCRA; Symbols and colours as in *Figure 1b*, symbol size corresponds to object size.

*Luppino, 2001*). Creating a motor plan and its execution is associated with areas F5 and M1 (*Murata et al., 1997*; *Raos et al., 2006*; *Umilta et al., 2007*).

In a first analysis that compared neuronal population tuning for the mixed (*Figure 3a*) and abstract objects (*Figure 3d*), we found evidence for a primary motor role of F5 and M1. The F5 population was strongly activated in the motor epochs when the mixed objects were grasped differently (up to 47% and 45% tuned neurons in monkey Z and M resp., see *Figure 3b*), and it was strikingly unmodulated when the abstract objects were grasped similar (*Figure 3e*). Likewise, the M1 population responded uniformly for similar grips (*Figure 3e*), but it was modulated very strongly when different hand configurations were applied (up to 67% and 61% in monkey Z an M resp, see *Figure 3b*).

During movement planning, M1 showed no or minimal preparation activity (*Figure 3a–c*), whereas F5 revealed a multimodal role: in the cue epoch, the tuned F5 population substantially decreased from 28% to 18% in monkey Z, and from 13% to 4% in monkey M, when comparing abstract with mixed objects. This is in strong contrast to AIP, which demonstrated equal population responses for both type of object sets. The reduced F5 modulation suggests early motor processes starting shortly after object presentation. However, 18% and 4% (Monkey Z and M resp.) of all F5 neurons remained their modulation when the monkeys observed the abstract geometries. Thus, at least some F5 cells coded objects in purely visual terms.

It is notable, that we found significantly different contributions in motor preparation between the F5 recording sites. The visual (*Figure 3e–f*) and visuomotor modulations (*Figure 3b–c*) prior to movement primarily originated from the ventral recording array, corresponding to the F5a subdivision (*Gerbella et al., 2011*; *Theys et al., 2012a*). In fact, 76% of the visual (abstract objects) and 72% of visuomotor tuned neurons (mixed objects) recorded from F5 were detected on the ventral site (ANOVA p<0.01, all sessions, tested in cue epoch), in line with previously reported enhanced decoding capabilities of planning signals from ventral F5 (*Schaffelhofer et al., 2015a*). In contrast, the dorsal F5 array, corresponding to F5p, mainly contributed during movement execution by a four-fold increase of its tuned population with respect to the cue epoch.

## Feature coding in area F5

The general population response of F5 was confirmed when extending our analysis to all 49 objects. Neurons were modulated by hand grasping actions and typically showed the strongest MD during motor execution. The example neuron shown in *Figure 4a–b* demonstrated a maximum MD of 39.8 Hz while grasping and allowed significantly separating 42% of all condition pairs in this epoch (*Figure 4b*, right). Importantly, the neuron showed similar motor coding already during the preparation epoch (r = 0.76, between plan and grasp MD-maps, as shown in *Figure 4b*). Using the planning activity of the example neuron alone, about 43% of the task conditions could be separated, thereby demonstrating the important role of F5 for hand movement planning.

To evaluate the relationship between neural population activity and motor actions, we recorded spiking activity (*Figure 4f–g*) together with the kinematics of the primate hand (*Figure 4c–d*). Dimensionality reduction methods (PCA, CDA) were performed to express the high dimensional kinematic (J-space) and neural space (N-space) in a low dimensional fashion. Procrustes analysis (PCRA) was subsequently applied to align the N- to the J-space for their visual comparison (*Figure 4e,h-i*) (see Materials and methods)

In detail, joint trajectories were recorded in 3-D space from an instrumented glove (*Schaffelhofer and Scherberger, 2012*) and translated to joint angles utilizing a primate-specific

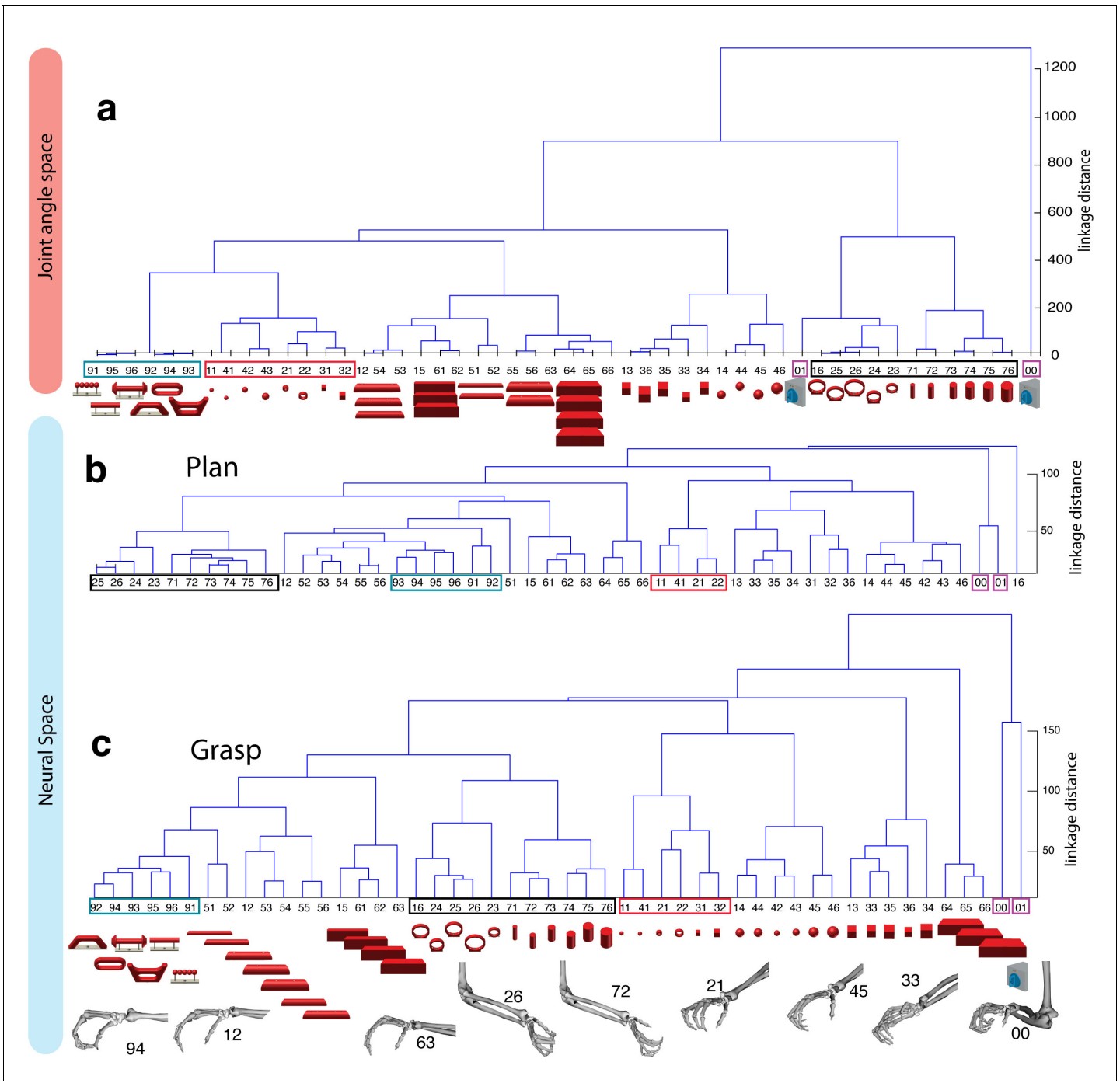

**Figure 5.** Hierarchical cluster analysis of the F5 population. (**a**) Dendrogram of J-space (27 DOF). (**b–c**) Dendrogram of F5's complete N-space during the plan and the grasp epoch. Condition numbers as in *Figure 1b*. A selection of grip types and their corresponding objects are illustrated. In a-c, similar motor characteristics are highlighted with coloured boxes (see text). (**b,c**) is based on the complete F5 population (N = 119, simultaneously recorded neurons), in contrast to its illustration in the reduced neural space in *Figure 4*. See *Figure 5—figure supplement 1–2* for the averaged population results across all sessions of animal M and animal Z, respectively.

The following figure supplements are available for figure 5:

**Figure supplement 1.** F5-Motor coding in animal M across all recording sessions.

**Figure supplement 2.** F5-Motor coding in animal Z across all sessions.

musculoskeletal model (*Schaffelhofer et al., 2015b*). The mean values of a total of 27 degrees of freedom were then extracted from the hold epoch to describe the hand shapes used for grasping the objects (J-space, *Figure 4c–d*). This provided accurate and stable, but highly variable hand configurations across the 49 tested objects. Performing PCA on this dataset allowed visualizing all correctly performed grips in a low-dimensional fashion. Thus, each marker in *Figure 4e* reflects one individual grip/trial.

Similarly, spiking activity from a large population of neurons was recorded with the FMAs. Then, mean firing rates were extracted from epochs of interest (e.g., grasp) as illustrated in *Figure 4f-g* (N-space). On this dataset we performed CDA and PRCA to reduce and compare the multidimensional N-space and the J-space as shown in *Figure 4h* and *Figure 4e*, respectively. For an animation of *Figure 4* see *Video 4*.

The J-space demonstrated a high variability of hand configurations across conditions and closely reflected the hand's wrist orientation (1st principal component) and hand aperture (2nd principal component). Furthermore, the reduced J-space allowed observing the most relevant kinematic observations with respect to the presented objects, as shown in *Figure 4e* and *5a*: (1) Objects of small sizes such as the small rings (condition ID 21, 22), spheres (11,41,42) and cubes (31, 32) were grasped similarly (index finger and thumb) and thus were located close to each other in the reduced and the complete J-space. (2) Vertical cylinders and big rings (16, 23–26, 71–76) were enclosed with the digits and required 90° of wrist rotation. Therefore, these grips are located close to each other in J-space. (3) All abstract objects (91–96) shared a compact cluster and demonstrated their similarity in J-space. (4) Precision (00) and power grips (01) performed on the same handle required highly different hand configurations and were located distant in J-space. (5) The highest separation of hand configuration across objects of similar shape was evoked from the rings that were grasped with precision (21, 22) or power grips (23–26). Small rings and big rings are therefore separated in the J-space.

Importantly, the majority of kinematic observations were also found in the N-space of F5 during motor execution (*Figure 4h*). In contrast to AIP, conditions of different visual stimuli but equal grips were located close to each other in the neural space (observation 1–3), whereas conditions of similar visual stimuli but different motor responses caused a separation (observation 4–5). These results were further supported by the high similarity between the J- (*Figure 4e*) and N-space (*Figure 4h*) during movement execution (for quantification see section 'Numerical comparison' and *Figure 7*). The findings demonstrate clearly different coding properties with respect to AIP and reveal a primary motor role of F5 during movement execution. CDA and PRCA were further used to visualize the evolution of the F5 population during the task. Similar to observations at the single unit level (see *Figure 4a–b*), the F5 population demonstrated first modulations and expressions of the upcoming motor actions already during motor preparation.

To quantify the observations made on the reduced spaces, hierarchical cluster analysis (HCA) was performed with the complete population of F5 neurons (N-space) and joint kinematics (J-space). In accordance with the low-dimensional representation, abstract forms, small objects, as well as the big rings and cylinders created individual clusters in the J-space and were located close to each other. As shown in *Figure 5b*, these motor characteristics were rudimentarily marked in F5 already during motor preparation (see *Figure 5b*, coloured boxes). These clusters that emerged during motor planning persisted to a large extent during motor execution, but increased their relative distance to each other (*Figure 5c*), in line with the higher MD of single F5 neurons during grasp execution (*Figure 4b*). HCA performed on the averaged population response across all recording sessions (see Materials and methods) confirmed in both animals the motor characteristics of simultaneously recorded populations in single sessions (*Figure 5—figure supplements 1–2*).

As discussed above, premotor preparation activity primarily originated from the ventral F5 array. Thus, the modulations observed prior to movement execution, such as observed in the CDA and hierarchical clustering, are mainly based on the ventral recording site. In contrast, both arrays significantly supported movement execution. The different modalities in both recording sites are in line with the architectonically (*Belmalih et al., 2009*) and connectionally (*Borra et al., 2010*; *Gerbella et al., 2011*) distinct subdivisions F5a and F5p, which largely correspond to the ventral and dorsal recording array, respectively. Distinct connections of F5a with SII, AIP and other subdivision of F5 (but not to M1) (*Gerbella et al., 2011*) suggest an integration of visual, motor and context specific information (*Theys et al., 2013*). On the other hand, connections of F5p to the hand area of M1

and to the spinal cord (*Borra et al., 2010*) suggest a rather direct contribution to hand movement control. Taken together, the multimodal preparation signals, including visual and motor contribution, and the distinct motor feature coding towards planning and execution, supports the key role of F5 for visuomotor transformation.

## Feature coding in area M1

In agreement with the general population response (*Figure 3b*), single neurons of M1 were almost exclusively modulated during movement execution and showed minimal modulations during preparatory epochs (*Umilta et al., 2007*), as also indicated by the example neuron in *Figure 6a–b*. This neuron was capable of differentiating 52% of condition pairs when holding the object, but failed to separate any (0%) of the conditions when planning the movement. (ANOVA, Tukey-Kramer criterion, p<0.01), thereby highlighting its major difference to F5 neurons.

Although the motor relevance of the hand area of M1 has been described extensively with electrophysiological (*Schieber, 1991*; *Schieber and Poliakov, 1998*; *Spinks et al., 2008*; *Umilta et al., 2007*) and anatomical methods (*Dum and Strick, 2005*; *Rathelot and Strick, 2009*), it has been unclear how versatile hand configurations are encoded at the population level. The high variability of objects and motor affordances in our task allowed such a description (*Figure 6*). Similar to the analysis performed on the F5 population, PCRA analysis compared the J-space with the N-space of M1 and revealed, highly important for the understanding of hand movement generation and as an important control, striking similarities between both representations, as demonstrated in *Figure 6c,e*.

The similarity of J- and N-space of M1 was not only visible in the first two components, but was also quantified across all dimensions in the hierarchical cluster trees of the simultaneously recorded M1 population (*Figure 6d,f*) and when averaging the population response across all recording sessions (*Figure 6—figure supplement 1–2* for monkey M and Z, respectively). The large majority of conditions were assigned to the same clusters in the J- and the M1-space (coloured boxes in *Figure 6d,f*, exceptions: conditions 42, 43). Furthermore, all of the motor characteristics (1–5) defined above were also observed and were even more strongly represented in M1 than in F5: again, the group of small objects (11, 21–22, 31–32, 41–42) and the group of abstract forms (91–96) created strong and individual clusters, whereas small (21–22) and big rings (16, 23–26) as well as precision (00) and power grips (01) were located distant from each other.

## Numerical comparison

We also quantified motor similarities between the J-space and the N-space of area AIP, F5, and M1 across all recording sessions, similar to individual recording session analyses presented in *Figures 4–6*. For this, similarity measures were performed between the J-space and the N-space of AIP, F5 and M1 using PCRA (see Materials and methods). A similarity of '0' indicates a complete match between the distribution of trials (n > 500) in N-space and in J-space, whereas values close to '1' represent high divergences between the clustering in both spaces. In accordance with the previous analysis, M1 demonstrated the highest similarity to the J-space (averaged value across 10 recording sessions for monkey M: 0.48, for monkey Z: 0.59), followed by F5 (monkey M: 0.61; monkey Z: 0.65) and AIP (monkey M: 0.75; monkey Z: 0.75). These values were highly consistent across recording sessions and monkeys (*Figure 7*). Furthermore, differences between areas were significant (ANOVA and post-hoc Tukey-Kramer criterion, p<0.01). Together, these results highlight the different roles of the cortical areas AIP, F5, and M1 for the preparation and execution of grasping movements.

## Feature code correlation

For visualizing the communication of features between areas, we correlated the dynamic modulations on a trial-basis (see Materials and methods). In this, the elicited modulation patterns of each neuronal population were correlated in a pair-wise fashion (AIP vs. F5, F5 vs. M1, AIP vs. M1) across time (*Figure 8*, *Video 5*).

During object presentation, we found similar chequered patterns in area AIP in both animals (t1 in *Figure 8*), again confirming the coding of object shapes at the population level, but here on trial-by-trial basis. These visual feature patterns caused a significant correlation peak with area F5 in animal Z, suggesting the propagation of visual information; in animal M, however, only a minor

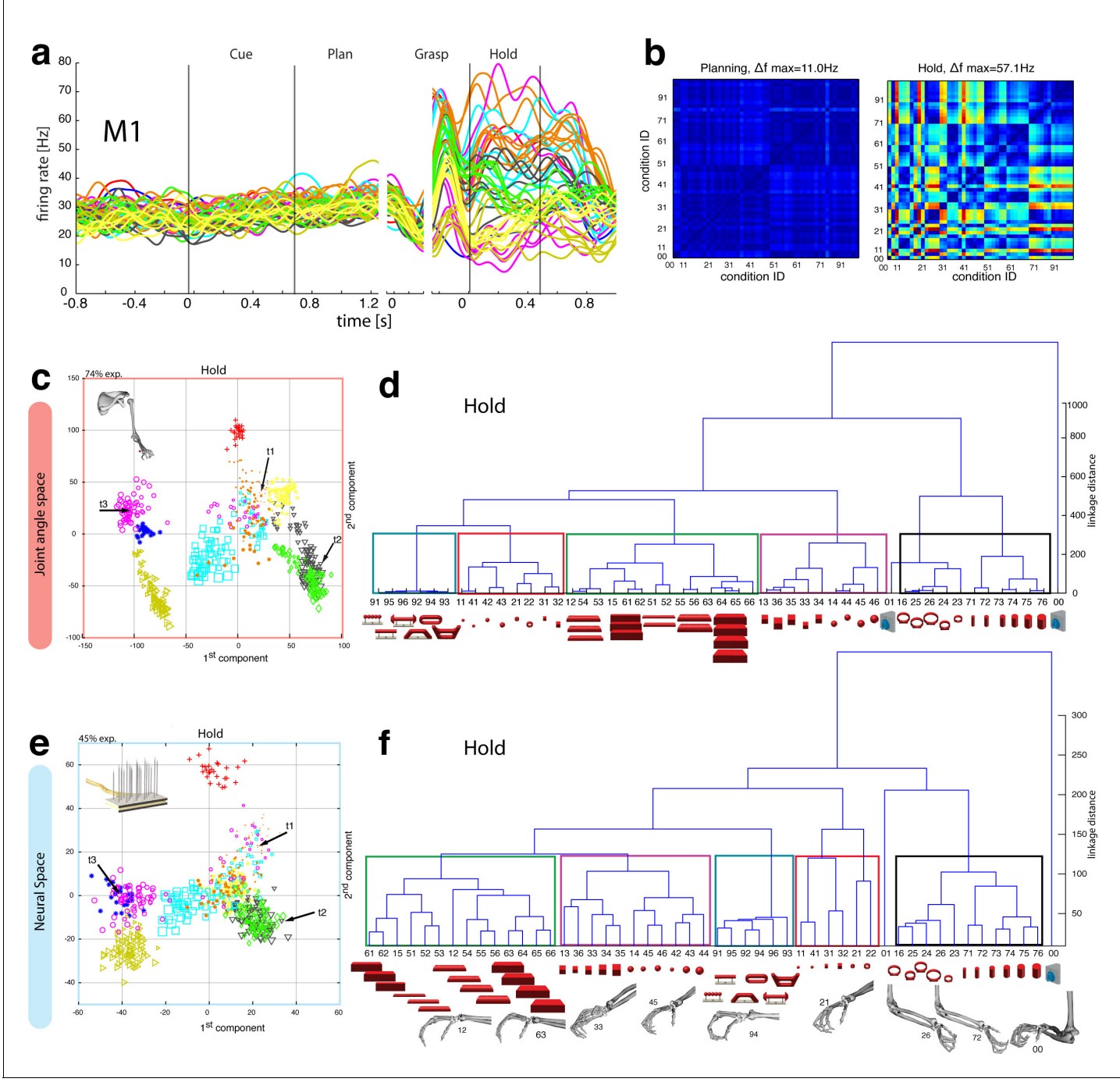

**Figure 6.** Motor execution in M1. (a) Example neuron of M1 in monkey M, curves show firing rates separately for all 50 task conditions (colour code as in **Figure 1b**). (b) Modulation depth plots (as in **Figure 4b**) for the planning and hold epoch of the example neuron in a. (c) Population activity from the reduced and (d) the complete kinematic space (J-space) is compared to (e) the reduced and (f) the complete neural population space (N-space) of M1 during the hold epoch of the task (N = 151, simultaneously recorded). In e, N-space was aligned to J-space with PCRA; symbols and colours as in **Figure 1b**; symbol size corresponds to object size. Arrows t1-t3 highlight the example trials of **Figure 4d,g**. See **Figure 6—figure supplement 1–2** for the averaged population results of animal M and animal Z, respectively.

The following figure supplements are available for figure 6:

**Figure supplement 1.** M1-Motor coding in animal M across all sessions.

**Figure supplement 2.** M1-Motor coding in animal Z across all sessions.

correlation peak was observed (*Figure 8*). These results are consistent with the different proportion of F5 visual cells identified in animal Z and M (*Figure 3*), which only temporarily shared the visual coding with AIP. Interestingly, F5 and AIP demonstrated minimal similarities during movement planning, in support of the different encoding schemes of visual and motor features described at the population level (*Figures 2*,*4* and *5*).

F5 modulations strongly increased before movement execution and were followed by M1 activity after movement onset (see *Video 5*), suggesting that premotor cortex drives M1. Signals did not only follow in time but also in modulation patterns, as expressed by the impressively high correlation coefficients in both animals (t2 in *Figure 8*), and in agreement with similar motor coding schemes observed with the population analysis. Together, these results expand the static feature coding found with CDA by demonstrating dynamical functional coupling of AIP, F5, and M1 during the course of the delayed grasp and hold task.

## Discussion

In this study, we took advantage of a rich set of objects and parallel recording techniques to study visuomotor transformation. We show that the cortical grasping network differentiates visual from motor processes to achieve this transformation. Area AIP revealed a strong visual role and a specialisation for processing object shapes. In contrast, area F5 coded objects only partially and temporarily in visual terms and switched to dominant motor signals towards movement planning and execution. The encoded motor features of F5 showed striking similarities with M1 thereby suggesting a strong collaboration of both areas during hand and movement control.

### Recording sites and relation to anatomical connections

We targeted the cortical grasping circuit with FMAs implanted under anatomical considerations. AIP, an end-stage area of the dorsal visual stream, receives input from parietal visual areas (e.g., LIP, CIP, and V6a) as well as from the inferior temporal cortex (e.g., areas TEa and TEm) (*Borra et al., 2008*; *Nakamura et al., 2001*). AIP further connects to pre-motor F5 via dense reciprocal projections (*Borra et al., 2008*; *Gerbella et al., 2011*; *Luppino et al., 1999*). In agreement with these known connections, we observed strong visuomotor responses in AIP and F5 (*Figure 3*).

Another significant connection links F5 with the hand area of M1 (*Dum and Strick, 2005*; *Kraskov et al., 2011*). As described by *Rathelot and Strick (2009)*, neurons of M1, in particular in the bank of the central sulcus, form direct connections to $\alpha$-motor neurons in the spinal cord that drive the distal hand muscles. In line with these observations, the majority of M1-neurons demonstrated significant modulations during hand movement control despite highly constant reaching

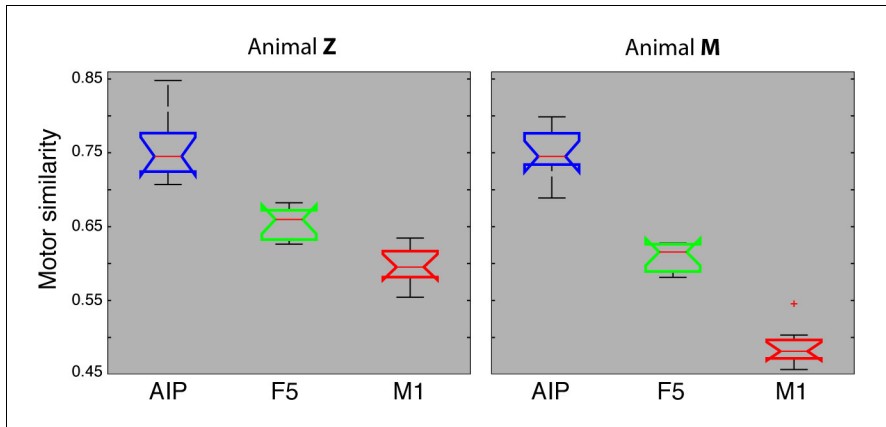

**Figure 7.** Motor similarity measure. Boxplots illustrating similarity between the population coding of J-space and N-space during the hold epoch of the task, as provided by the PCRA analysis. Left: results in AIP, F5, and M1 across all 10 recording sessions for monkey Z. Right: same for monkey M. Red horizontal lines indicate median value, boxes show lower and upper quartile of data (25%–75%), and whiskers indicate maximum and minimal values.

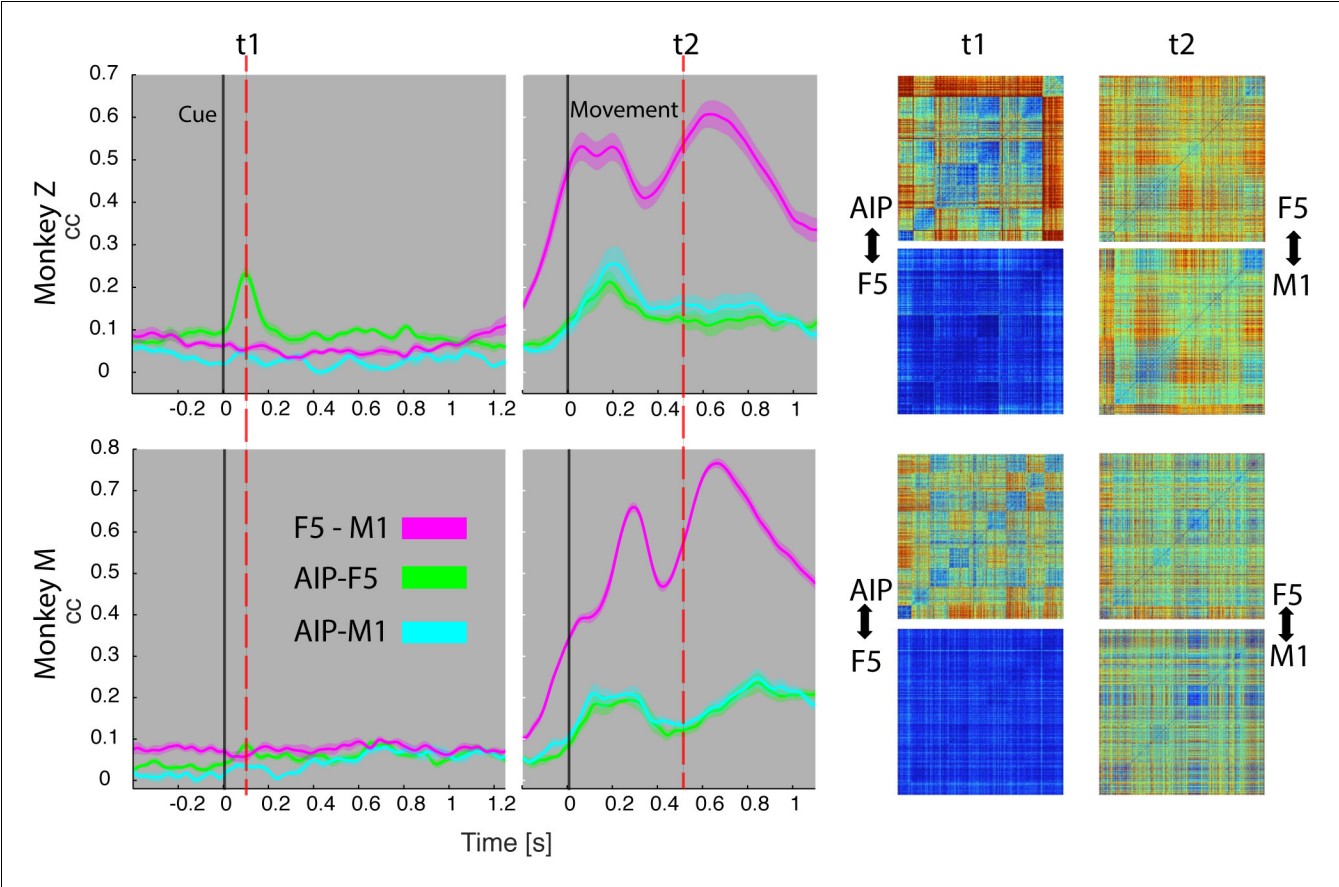

**Figure 8.** Temporal feature correlation between areas. The distance (in firing rate) between all possible trial-pairs was computed separately in the N-space of area AIP, F5 and M1. The resulting distance maps thus represent the neuronal modulations of a population (e.g. AIP) for a specific time t. (Right) Example maps show the neural modulation patterns at key times t1 (object presentation) and t2 (during hold). Warm colours: long neural distances, cool colours: short neural distances. (Left) Correlating the neuronal patterns across time (Spearman's r) allowed visualizing the similarity between the areas for animal Z (top row) and animal M (bottom row). For computation, spiking activity was aligned to the beginning of the cue and grasping onset. For an animation of the feature code correlation see *Video 5*.

components (*Figure 3b*). Together, strong grasp modulations (*Figure 3*) and their intercommunication (*Figure 8*) confirmed the correct positioning of the electrode arrays and the importance of all areas for visuomotor processing.

## Visual processing for grasping

While it is known that area AIP and F5 strongly respond to the presentation of objects and to the variation of their dimensions (e.g. shape, orientation) (*Baumann et al., 2009*; *Fagg and Arbib, 1998*; *Murata et al., 1997*; *2000*; *Taira et al., 1990*) it was unclear whether these modulations reflect visual or motor processes. For example, responses to object presentation can either reflect object attributes or instant motor representations as we have shown here (*Figure 3*) and as suggested previously (*Murata et al., 1997*). By design, our task created associations as well as dissociations between objects and their afforded hand configurations (*Figure 4e*, *5a*) in order to disentangle these normally highly linked parameters.

The AIP population demonstrated a distinct visual separation of objects (*Figure 2*, *3e*), which was largely unrelated to the observed motor characteristics (*Figure 4e*, *5a*), even when grasping in complete darkness (*Figure 7*). The predominant criterion for object separation was object shape, even for abstract geometries that required the same grip (*Figure 3d–f*). These findings are in agreement with anatomical connections of AIP to the inferior temporal cortex (*Borra et al., 2008*) that codes perceived shapes (*Logothetis et al., 1995*; *Tanaka, 1996*). Object size was expressed in the

population but played, surprisingly, only a secondary role in AIP (*Figure 2*). These smaller neural distances with respect to size are remarkable, in particular since this feature is highly relevant for controlling hand aperture (*Jakobson and Goodale, 1991*). A possible explanation for this effect could be the higher computational effort (more neurons) required for processing shape in comparison to size.

AIP also widely maintained its coding for visual object properties during movement execution in the dark (*Figure 2f*), although with significantly smaller neural distances (*Figure 2e–f*). We hypothesize that AIP serves as working memory and not only extracts, but also maintains visual object information required for motor planning and execution (*Rizzolatti and Luppino, 2001*). We emphasize that visual coding at the population level does not preclude motor coding of some individual cells, as suggested previously (*Murata et al., 1997*). Rather, the evidence of bidirectional connections to F5 suggests that subpopulations exist in AIP that reflect feedback of motor signals.

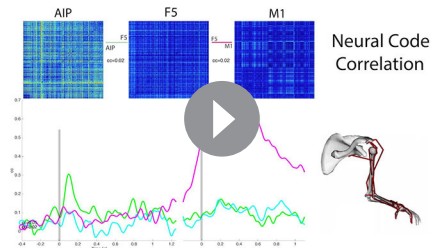

**Video 5.** Temporal feature correlation between areas. The neural distance (in firing rate) between all pairs of trials within a neuronal population provided a modulation pattern between trials (top) for area AIP, F5, and M1. Correlating these neuronal patterns for each moment in the task allowed visualizing the coding similarity between pairs of areas across time (AIP-F5 in green, AIP-M1 in cyan, and F5-M1 in magenta). Feature correlations are based on all trials, whereas the presented object and corresponding grasp movement are shown for an example trial.

Despite AIP's primary visual coding, we observed a distinctive coding in the N-space when the same objects were grasped differently (e.g., handle and cylinders in *Figure 2c–d*). These modulations could reflect either motor (*Fagg and Arbib, 1998*) or visual transformations required for grasping. In fact, when grasping an object, both processes are required: the visual selection of (focus on) object parts and the selection of a corresponding hand configuration. Several indicators in our data suggest visual rather than motor transformations in AIP: First, the population separated the entire set of objects in visual terms, but differentiated same objects when grasped differently. Second, the separation of the same object when grasped differently, originated from one object shape cluster. In line with these findings, *Baumann et al. (2009)* demonstrated that AIP planning activity depends on the previous visual knowledge of an object (see *Figure 6* in their paper). Presenting an object (the same handle as we used) caused a strong visual activation in the AIP population that got separated when a grip type (precision or power grip) was instructed subsequently. In contrast, instructing the grip type before object presentation did not lead to a substantial population response and caused significant modulations only after the object was presented. These differentiation schemes support our hypothesis of visual rather than motor transformations in AIP.

## Motor planning and execution

Grasping requires the transformation of visual descriptions of an object into adequate motor commands. F5 is densely connected to AIP (*Borra et al., 2008*; *Gerbella et al., 2011*; *Luppino et al., 1999*) and has been associated with these visuomotor processes (*Fluet et al., 2010*; *Jeannerod et al., 1995*; *Rizzolatti and Luppino, 2001*). Similar to AIP, neurons in F5 respond to the presentation of 3D objects (*Fluet et al., 2010*; *Murata et al., 1997*; *Raos et al., 2006*; *Theys et al., 2012a*; *Vargas-Irwin et al., 2015*). These modulations have been discussed to reflect object or motor representations (*Fluet et al., 2010*; *Murata et al., 1997*; *Raos et al., 2006*; *Theys et al., 2013*), however without measuring corresponding hand kinematics.

With many more object conditions, precise hand kinematics, and a large neuronal dataset we were able to address this question and confirm a primary motor role of the F5 population for representing the joint (J-) space during motor execution (*Figures 4*, *5*). Due to the large number of objects tested, we could demonstrate for the first time that the F5 population does not reflect stereotypical grip types (*Rizzolatti and Luppino, 2001*), but represents a continuum of many hand configurations (*Figure 5*), highly similar to M1 (*Figure 8*, *Video 5*).

Importantly, motor characteristics in the N-Space of F5 could be observed not only in motor execution epochs, but also during motor preparation (*Figure 4i, 5b*). In contrast to AIP, the F5

population showed already in the cue epoch a reduced tuning for abstract objects requiring the same grip (*Figure 3a–b* vs. *Figure 3d–e*). This suggests a rapid appearance of grip features in F5 shortly after object presentation. However, F5 also contained a small subgroup of neurons that represented pure visual information during the cue epoch (*Figure 3*). The vast majority of these cells have been recorded from the ventral array, i.e., they originate from the F5a subdivision. In line, electrophysiological investigations of area F5a demonstrated selectivity for 3D shape (*Theys et al., 2012a*). We suggest that these neurons receive direct visual input from AIP and might be crucially involved in the activation of stored motor plans. Similar modulation patterns in AIP and F5 during object presentation suggest the communication of visual properties to premotor cortex (*Figure 8* and *Video 5*). The presence of multimodal signals as well as the proximity of specialized sub areas for visual input and motor output suggest a central role of F5 for visuomotor transformation.

In contrast to F5, the hand area of primary motor cortex (M1) showed an almost exclusive role for motor execution (*Saleh et al., 2010*; *Umilta et al., 2007*). We investigated, for the first time, the N-space of the hand area of M1 (*Rathelot and Strick, 2009*) for a large repertoire of grasping actions. As expected, the N-space of M1 was closely related to the J-space (*Figure 6*) and showed the highest motor similarity across all three areas (*Figure 7*).

Correlating these M1 modulation patterns with F5 revealed that both areas are not only strongly interconnected (*Dum and Strick, 2005*; *Kraskov et al., 2011*), but they also heavily share motor features for grasping (*Figure 8*). The high similarity of the F5 and M1 population during movement execution (*Figure 4h* vs. *Figure 6e*, *Figure 8*, and *Video 5*) is in agreement with findings of similar coding schemes between both areas in spiking (*Umilta et al., 2007*) and beta-band LFP activity (*Spinks et al., 2008*). The earlier premotor (vs. motor) onset (*Figure 3b*) suggests that the movement is facilitated by F5, whereas the closer motor similarity of M1 might reflect its advanced finger control capacity due to more direct motor connections (*Fogassi et al., 2001*; *Rathelot and Strick, 2009*; *Schieber and Poliakov, 1998*).

## Conclusions

Our findings demonstrate that the cortical grasping network transforms visual object attributes into motor plans and actions. We found highly different coding schemes between the frontoparietal circuits of AIP and F5, indicating widely separated processing of visual and motor features. These findings suggest that visuomotor transformation is achieved effectively by visual object descriptions that activate linked motor plans in the reciprocal network of AIP and F5. Strong feature communication between F5 and M1 further suggest a common motor network that executes and refines the prepared motor plan.

## Materials and methods

### Animal training and experimental setup

Two rhesus monkeys (*Macaca mulatta*) participated in this study (animal Z: female, 7.0 kg body weight; animal M: male, 10.5 kg). Animal housing, care, and all experimental procedures were conducted in accordance with German and European laws governing animal welfare and were in agreement with the guidelines for the care and use of mammals in neuroscience and behavioural research (*National Research Council, 2003*; see also Ethics statement).

We developed an experimental setup that allowed us to present a large number of graspable objects to the monkeys while monitoring their behaviour, neural activity and hand kinematics. During each recording session, monkeys grasped a total of 42–48 objects of equal weight that were placed on 8 interchangeable turntables (*Figure 1a–b*). Objects were of different shapes and sizes including *rings* (diameter of 15, 20, 25, 30, 35, and 40 mm), *cubes* (length of 15, 20, 25, 30, 35, and 40 mm), *spheres* (diameter of 15, 20, 25, 30, 35 and 40 mm), *horizontal cylinders* (diameter of 15, 20, 25, 30, 35, 40 mm, equal length), *vertical cylinders* (diameter: 15, 20, 25, 30, 35, 40 mm, equal height), and *bars* (depth of 15, 20, 25, 30, 35, and 40 mm, equal height and width). Furthermore, a *mixed* turntable held objects of different shapes of average size. Important for this study, a special turntable was holding objects of *abstract forms*, which differed largely visually but required identical hand configurations for grasping (*Figure 1b*). Both monkeys were also trained to grasp a single object, a handle, either with a precision grip or a power grip. This extended our task by two more conditions (to a

total of 50) that evoked similar visual, but different motor responses. In 20 recording sessions (10 per animal), each condition was repeated at least 10 times.

Precision and power grips applied on the handle as well as grasping and lifting the 3D-objects were detected with photoelectric barriers (*Schaffelhofer et al., 2015a*). The turntable position was controlled with a step motor. Furthermore, the monkey's eye position was monitored with an optical eye tracking system (model AA-EL-200; ISCAN Inc.). All behavioural and task-relevant parameters were controlled using a custom-written control software implemented in LabVIEW Realtime (National Instruments, Austin, TX).

## Task paradigm

Monkeys were trained to grasp 49 objects (*Figure 1b*) in a delayed grasp-and-hold task (*Figure 1d*). While sitting in the dark the monkeys could initiate a trial (self-paced) by placing their grasping hand (left hand in monkey Z, right hand in monkey M) onto a rest sensor that enabled a fixation LED close to the object. Looking at (fixating) this spot for a variable time (fixation epoch, duration: 500–800 ms) activated a spot light that illuminated the graspable object (cue epoch: 700 ms). After the light was turned off the monkeys had to withhold movement execution (planning epoch: 600–1000 ms) until the fixation LED blinked for 100 ms. After this, the monkeys released the rest sensor, reached for and grasped the object (movement epoch), and briefly lifted it up (hold epoch: 500 ms). The monkeys had to fixate the LED throughout the task (max. deviation: ~5 deg of visual angle). In trials where the handle was grasped, one of two additional LEDs was presented during the cue epoch, which indicated to perform either a precision grip (yellow LED) or a power grip (green LED). All correctly executed trials were rewarded with a liquid reward (juice) and monkeys could initiate the next trial after a short delay. Error trials were immediately aborted without reward and excluded from analysis.

## Kinematics recording

Finger, hand, and arm kinematics of the acting hand were tracked with an instrumented glove for small primates (*Figure 1a,c*). Seven magnetic sensor coils (model WAVE, Northern Digital) were placed onto the fingernails, the hand's dorsum as well as the wrist to compute the centres of 18 individual joints in 3D space, including thumb, digits, wrist, elbow and shoulder. The method and its underlying computational model have been described previously (*Schaffelhofer and Scherberger, 2012*).

Recorded joint trajectories were then used to drive a 3D-musculoskeletal model (*Schaffelhofer et al., 2015b*), which was adjusted to the specific anatomy of each monkey. The model was implemented in OpenSim (*Delp et al., 2007*) and allowed extracting a total of 27 DOF [see *Schaffelhofer et al. (2015b)* for detailed list of DOF]. All extracted joint angles from the model were low-pass filtered (Kaiser window, finite impulse response filter, passband cutoff: 5–20 Hz), downsampled to 50 Hz and used to describe the hand configuration in a 27-dimensional joint space (J-space).

## Electrophysiological recordings

Single and multiunit activity was recorded simultaneously using floating microelectrode arrays (FMA, Microprobe Inc., Gaithersburg, MD, USA). In each monkey we recorded from in total 192 channels of 6 individual arrays implanted into the cortical areas AIP, F5, and M1 (see *Figure 1e–g*). In each array, the lengths of the electrodes increased towards the sulcus and ranged from 1.5 (1st row) to 7.1 mm (4th row). In area F5, one array was placed in the posterior bank of the inferior arcuate sulcus approximately targeting F5a (longer electrodes) (*Theys et al., 2012a*) and approaching the F5 convexity (F5c; shorter electrodes). The second and more dorsally located array was positioned to target F5p. In AIP, the arrays were implanted into the end of the posterior intraparietal sulcus at the level of area PF and more dorsally at the level of area PFG. In M1, both arrays were placed into the hand area of M1 into the anterior bank of the central sulcus at the level of the spur of the arcuate sulcus (*Rathelot and Strick, 2009*). See *Schaffelhofer et al. (2015a)* for details on surgical procedures. Neural activity was recorded at full bandwidth with a sampling frequency of 24 kHz and a resolution of 16 bits (model: RZ2 Biosignal Processor; Tucker Davis Technologies, FL, USA). Neural data was synchronously stored to disk together with the behavioural and kinematic data. Raw recordings were

filtered offline (bandpass cutoff: 0.3—7 kHz) before spikes were detected (threshold: 3.5x std) and extracted. Spike sorting was processed in two steps: First, we applied super-paramagnetic clustering (*Quiroga et al., 2004*) and then revised the results by visual inspection using Offline sorter (Plexon, TX, USA) to detect and remove neuronal drift and artefacts. No other pre-selection was applied and single and multiunit activity have been analysed together.

## Data analysis

### Firing rate and modulation depth

Firing rate plots were created in order to visualize the activity of example neurons across time. For this, spike rates were smoothed with a Gaussian Kernel (σ = 50 ms) over time and then averaged across trials of the same condition (*Baumann et al., 2009*). To illustrate the response of a specific condition we used the colour code as introduced in *Figure 1b*. Furthermore, we visualized the modulation depth (MD) of example neurons in specific epochs of interest. The MD between two conditions was defined as their absolute difference in average firing rate. This measure was computed for all possible pairs of conditions and visualized as a colour-map. In addition, we performed multi-comparison tests to check whether the differences in firing rate between conditions were significant (ANOVA, Tukey-Kramer criterion, $p < 0.01$; $\geq 10$ trials/condition, $\geq 500$ trials/session, Matlab functions: anova1, multcompare).

### Sliding ANOVA population analysis

To investigate the population activity during the course of the trial, we tested for significant tuning at multiple time points t using a 1-way ANOVA. Similar to the visualization of firing rates, we first smoothed all spike trains with a Gaussian kernel (σ = 50 ms) and then performed a sliding ANOVA in time steps of 1 ms ($p < 0.01$; no multi-comparison correction, $\geq 10$ trials/condition). Sliding ANOVA results were then averaged across all recording sessions to visualize results (*Figure 3*). Due to the variable length of the planning epoch, trials were first aligned to the cue-onset and then to the grasp-onset event. Abstract shapes were introduced at a later stage of the project (included in 12 recording sessions, *Figure 3e*), whereas all other conditions were tested in all 20 recording sessions (*Figure 3b*). Please note: Irregular grips were detected in the J-space and excluded for the sliding ANOVA analysis of the abstract shapes (*Figure 3b*). Only 17 trials in 12 recording sessions were excluded to guarantee equal hand configurations for statistical analysis.

To calculate the visual response times of an area of interest (i.e. AIP and F5) we measured the time between cue onset and 75% of the population's peak activity. Response times were averaged across recording sessions for each animal and tested for significant differences.

### Dimensionality reduction

To compare and visualize the simultaneously recorded high-dimensional data (J- and N-space), we applied dimensionality reduction methods. We used principal component analysis (PCA) to express the data variability of the J-space in a compact, lower-dimensional fashion. This approach was demonstrated to be optimal for describing hand shapes (*Pesyna et al., 2011*). As an input for PCA, we computed for each trial and DOF the mean joint angle during the hold epoch (in degrees). This let for each trial to a 27-dimensional joint position vector in J-space that robustly described the monkey's average hand configuration during the hold epoch, and across all trials to an input matrix of dimension: trials x DOF.

For exploring the N-space (neuronal population space), we applied canonical discriminant analysis (CDA). Similar to PCA, CDA creates a new transform of the original dataset spanned by linear combinations of the original variables. Whereas PCA creates the new coordinate system in a way that maximizes the total variance, CDA transforms the data in order to maximize the separation of groups (here: task conditions). The first axis in the new transform (first canonical variable) therefore reflects the linear combination of original variables that show the most significant F-statistic in a one-way analysis of variance. The second canonical variable is orthogonal to the first one and has maximum separation within the remaining dimensions.

We performed CDA based on the population's mean firing rates of a specific task epoch of interest (e.g., grasp epoch) relative to baseline activity. This means, a population matrix consists of entries that correspond to the mean firing rate of individual neurons for all trials in a certain time

interval (dimensions: trials x number of neurons). In contrast to other dimensionality reduction methods, CDA considers only variances of the signal related to conditional differences. Due to this advantage, noise or condition-irrelevant modulations get suppressed.

For a fair comparison, all dimensionality reduction results presented for the individual areas originated from the same recording session per monkey (e.g. AIP, F5 and M1 populations shown from monkey Z or M) and were therefore recorded simultaneously.

## Procrustes analysis

Procrustes analysis (PCRA) can be used to test similarities or differences between multidimensional datasets of different measures and scales. In the context of this study, PCRA was used to evaluate the resemblance between the J- and the N-space (firing rates). For this, we reduced both representations to their highest common dimension (i.e., 27 DOF of hand and arm) using the dimensionality reduction methods explained above (i.e., CDA for N-space). This produced datasets of identical number of trials (e.g., 600) and dimensions (e.g., 27). Then PCRA was used to translate and rotate the space of interest (e.g., N-space of M1, F5, or AIP) in a way that minimized the sum of squared distances (SSD) to the corresponding points in the reference space (e.g., J-space) (Matlab function: procrustes). The resulting transform was used to visualize and quantify the amount of similarity to the reference space (J-space). As a numerical measure of similarity, the sum of squared distances between the new transform and the reference frame was computed and normalized by the sum of squared distances between points of the reference space to their dimensional means (Matlab function: procrustes). This similarity measure is a non-negative number with values near 0 implying a high similarity between the multidimensional spaces and values approaching (or exceeding) 1 implying strong dissimilarity.

## Hierarchical cluster analysis

To illustrate and compare the many conditions of our task in an untransformed, full-dimensional way, we performed hierarchical cluster analysis (HCA). We first computed the Mahalanobis distances between the population activities (N-space) of all possible pairs of task conditions in an epoch of interest (Matlab function: manova1). This resulting distance matrix (e.g., for 50 x 50 conditions) was used to create an agglomerative hierarchical cluster tree based on the average linkage criterion (Matlab function: manovacluster), presenting the cluster solutions of an individual recording session as dendrograms (e.g. *Figure 2e–f* for AIP of animal Z). Additionally, we averaged the Mahalanobis distance matrices of all individual session to build dendrograms expressing the N-space of an animal across all recording sessions (e.g. *Figure 2—figure supplement 1* for AIP of animal Z).

## Feature code correlation

For correlating the dynamic feature communicated within the grasping network we (1) binned the spiking activity of each individual neuron (bin width = 10 ms), (2) smoothed the firing rates with a Gaussian kernel ($\sigma$ = 300 ms) and (3) aligned the firing vectors to both cue and grasp onset. Within each of the resulting bins we then (4) computed the Euclidean distance between all possible pairs of trials in the N-space. This leads, separately for each area AIP, F5 and M1, to a distance matrix of the size $m^2$, where m is the number of trials that represents the neural population difference across different trials and trial conditions. Pairs of such matrices from AIP, F5, and M1 were then (5) correlated for every time bin (Spearman's correlation coefficient), which led to a correlation function across time for each pair of areas (AIP-F5, AIP-M1, and F5-M1). This correlation function represents for every time in the task the similarity of the encoded features between both areas. *Figure 8* and *Video 5* visualizes these distance matrices and the resulting correlation function over the time.

## Acknowledgements

The authors thank M Sartori, M Dörge, K Menz, R Ahlert, N Nazarenus, and L Burchardt for assistance and MM Fabiszak, M Hepp-Reymond, and W Freiwald for helpful comments on an earlier version of the manuscript. This work was supported by the Federal Ministry of Education and Research (Bernstein Center for Computational Neuroscience II Grant DPZ-01GQ1005C, the German Research

Foundation (SCHE 1575/3-1), the European Union Grant FP7– 611687 (NEBIAS), and the Humboldt Foundation.

## Additional information

### Funding

| Funder | Grant reference number | Author |
|---|---|---|
| Alexander von Humboldt-Stiftung | Postdoctoral Fellowship | Stefan Schaffelhofer |
| Deutsche Forschungsgemeinschaft | SCHE 1575/3-1 | Hansjörg Scherberger |
| Bundesministerium für Bildung und Forschung | BCCN-II, 01GQ1005C | Hansjörg Scherberger |
| European Commission | FP7-611687 (NEBIAS) | Hansjörg Scherberger |

The funders had no role in study design, data collection and interpretation, or the decision to submit the work for publication.

### Author contributions

SS, Designed experiments, Developed the experimental setup, Trained the animals, Recorded and analysed the data, Wrote the manuscript, Acquisition of data; HS, Designed experiments, Performed the surgeries, Edited the manuscript, Provided supervision at all stages of the project, Analysis and interpretation of data

### Author ORCIDs

Stefan Schaffelhofer, http://orcid.org/0000-0002-1006-971X
Hansjörg Scherberger, http://orcid.org/0000-0001-6593-2800

### Ethics

Animal experimentation: All procedures and animal were conducted in accordance with the guidelines for the care and use of mammals in neuroscience and behavioral research (National Research Council, 2003), and were in agreement with German and European laws governing animal care. Authorization for conducting this study has been granted by the regional government office, the Animal Welfare Division of the Office for Consumer Protection and Food Safety of the State of Lower Saxony, Germany (permit no. 032/09). Monkey handling also followed the recommendations of the Weatherall Report of good animal practice. Animals were pairhoused in a spacious cage (well exceeding legal requirements) and were maintained on a 12-hour on/off lighting schedule. Housing procedures included an environmental enrichment program with access to toys, swings, and hidden treats (e.g., seeds in sawdust). Monkeys had visual and auditory contact to other monkeys. They were fed on a diet of enriched biscuits and fruits. Daily access to fluids was controlled during training and experimental periods to promote behavioral motivation. All surgical procedures were performed under anesthesia, and all efforts were made to minimize post-surgical pain or suffering. Institutional veterinarians continually monitored animal health and well-being.

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
