## [Decision Letter]

Thank you for submitting your article "Object vision to hand action in macaque parietal, motor, and premotor cortices" for consideration by *eLife*. Your article has been reviewed by three peer reviewers. The following individuals involved in the review of your submission have agreed to reveal their identity: Kenneth Valyear (peer reviewer) and Sabine Kastner (Senior Editor and peer reviewer).

The reviewers have discussed the reviews with one another and the Editor has drafted this decision to help you prepare a revised submission.

Summary:

All three reviewers felt that this was an exceptionally strong study providing fundamental insight into visuo-motor transformations during grasping actions.

Essential revisions:

The reviewers thought that all comments deserve consideration by the authors. Particularly, the reviewers felt that analyzing the data as a function of the F5 subdivisions would strengthen the study further, particularly since the authors have followed such a strategy in previous work, and we'd hope that such analysis will be straightforward. In addition, the reviewers felt that there were interpretational issues with the role of area AIP that will deserve careful attention.

Reviewer #1:

The study by Schaffelhofer and Scherberger on "Object vision to hand action in macaque parietal, motor, and premotor cortices" reports novel findings on the roles of areas AiP, F5 and M1 in the transformation of visual information to motor action. Simultaneous recordings in these areas from arrays of electrodes were performed. The animals were trained on an excellent grasping task that used a variety of 3D objects and dissociated the different parts of a grasping task into its component parts (e.g. visual, motor planning, motor action). In addition to the use of sophisticated behavioral and recording protocols, the authors also used a glove to quantitatively measure kinematic aspects of grasping movements. The complex data from multiple areas and recording channels as well as movement parameters were analyzed in sophisticated ways to permit novel insight into the transformation of visual signals in areas AIP and F5 to motor planning and action in F5 and M1. I'll leave the detailed critique to the expert reviewer. In focusing on a larger picture in this particular area, I found this paper outstanding in all respects.

Reviewer #2:

1) The main limit of this beautiful work is the assumption that area F5 is anatomo-functionally unitary. Recent literature not considered in this work shows that F5 is neither architectonically (Belmalih et al. 2009) nor connectionally (Gerbella et al. 2011) homogeneous: the ventral and anterior part of F5 (called F5a, where the most lateral arrays in the present study have been implanted), hosts 3D-shape-selective and "visual-dominant" neurons (Theys et al. 2012), commonly found also in AIP (Theys et al. 2013) but not in the dorsal part of F5 (i.e. F5p, see Bonini et al. 2014). Furthermore, while in the introduction the authors generally talk about "connections of F5 to the spinal cord and M1", this is true virtually only for F5p (Borra et al. 2010). Considering altogether data recorded from different F5 subdivisions was acceptable in the authors' previous paper (Schaffelhofer et al. 2015, J Neurosci), as its goal was to decode hand configurations, but not here, where the authors want to elucidate the functioning of the cortical grasping network.

2) Strictly related to this issue, the electrodes length is unknown. From the authors' previous works (Schaffelhoefer et al. 2015a and Townsend et al. 2011) I infer that "electrode length ranged from 1.5 to 7.1 mm", with the longest electrodes toward the sulcus. Thus, the more superficial electrodes of the lateral arrays certainly sampled F5c motor activity, likely related to mouth and hand movements, (see Graziano et al. 1997 and Maranesi et al. 2012), while the deepest electrodes most likely sampled F5p/a visuomotor activity. Please clarify.

3) "Data have been analysed from 20 recording sessions (10 per animal)…". I don't understand how many (single and multi) units have been used for the analyses. It seems that a total of about 202 (units)*10 (sessions)=2020 and 355*10=3550 "units" have been considered in the two animals, respectively: with chronic recording this approach would certainly imply some resampling of the same "units" across sessions, increasing their number but reducing the variability of their discharge. At the same time, the authors stated in the Materials and methods section that "all dimensionality reduction results presented for the individual areas originated from the same recording session per monkey", which actually sounds fair, but in contrast with the previous consideration. Please clarify.

4) I recommend using the same y-axis scales for plots in Figure 3). Nevertheless, it is clear that the tuning profile of AIP does not change when abstract or mixed sets of objects are used (see Figure 3): I totally understand the authors' reasoning, but they will acknowledge that all the abstract objects share an elongated shape, while those of the mixed set are much more variable. Thus, the lack of difference in the AIP tuning profile between the two sets is strange and should be better explained/discussed. For example, could it be due to receptive fields properties of AIP neurons (see Romero et al. 2014; Romero and Janssen 2016)?

Reviewer #3:

Altogether, I consider the work exceptional. The data make new and important contributions to understanding the cortical mechanisms underlying grasp control. By concurrently recording and characterising the relationships between limb kinematics and neural activity from AIP, F5, and M1 while monkeys viewed and grasped a wide range of objects, the scope of the current investigation is unparalleled in the current primate neurophysiological literature. Also outstanding are the beautiful data visualisation methods that the authors advance here.

I have a few suggestions, however, detailed below. My hope is that the authors will find these comments constructive.

A) Main concerns:

(A-1) The role of AIP

I'm having some difficulty understanding the authors’ stance on the role of AIP. I believe they are arguing that their data suggest that AIP is encoding object properties in visual terms, and not in motoric terms. They suggest that the results of their CDA show that AIP encodes objects according to shape in visual space, independent of the way they are grasped.

My interpretation of their stance is that AIP activity does not reflect action possibilities in sensorimotor terms – but rather reflects the 'flagging' of visual features relevant for actions, but NOT (yet) represented as spatiomotor plans.

If this is indeed their stance, I see this as a significant departure from prior conceptualizations (Jeannerod et al., 1995).

However, I remain confused about this argument, both (1) conceptually, and (2) with respect to how some aspects of their data can be viewed as consistent these claims.

1) Conceptually. The authors talk about "object affordances" as different from "motor affordances".

To me, affordances = action possibilities; they are sensory representations of the external environment that are inseparably linked to the motor apparatus and capabilities of the perceiving animal. In other words, affordances are, according to my understanding, inextricably motor.

What does it mean to "extract object affordances relevant for grasping", but to not at the same time encode those same objects according to the spatiomotor properties required for their manual interaction?

They state: "…suggest encoding of object affordances rather than motor affordances".

I urge the authors to better clarify what is meant by object vs. motor affordances.

2) Data. Some aspects of AIP data suggest encoding related to motoric features.

a) Shift for same object – big horizontal cylinders – when grasped from below vs. above.

b) Significant differentiation of conditions 00 vs. 01.

c) Motor similarity measures – Figure 7 – presumably a measure of.75 for AIP is different than a measure of "1" – and thus, does suggest some correspondence between N- and J-space encoding in AIP

d) Also, there interpretation of Figure 3 vs. d – "…observed encoding of abstract objects in AIP can be explained exclusively by object shape."

Is it not possible that these data may reflect the fact that abstract versus mixed object set have more graspable parts – a greater number of possible actions –, and AIP encoding is sensitive to these differences?

This account could be considered consistent with the authors’ interpretations regarding the evidence for reduced tuning in AIP over time for abstract shapes: "…reduction of 3D shapes to the relevant parts for grasping…".

Summary:

While the authors seem to argue for a strict separation in roles between AIP and F5 – visual vs. motor, respectively – I see the data as more likely evidence for a continuum – a graded difference – visual prevalence in AIP, with motor encoding vs. motoric prevalence in F5, with visual encoding.

I suggest that the authors clarify their stance, and/or temper their arguments.

B) Intermediate concerns

B-1) in the Introduction, the authors state that prior work did not dissociate between visual and motor features. At first, I thought this was inconsistent with the work of Sakata and colleagues – describing visual dominant, visuomotor, and motor dominant cell types. I think some discussion of this work upfront, including how the current approach critically differs, would be helpful.

B-2) Introduction – last paragraph – it isn't clear to me what the authors mean to say here, without first reading the rest of the paper. I think this section could be made clearer.

B-3) I'm not sure I understand their explanation of the shape-specificity prior to fixation. Please try to make this clearer.

B-4) At first I thought I missed something – it seemed as though AIP was not looked at in terms of J- and N-space encoding in the same way that F5 and M1 were. However, later it is clear that they did analyse AIP in the same way. Why not also show these data, as they do for F5 (Figure 4, Figure 5) and M1 (Figure 6)?

B-5) Figure 4 – it is unclear to me why they plot J-space for the hold epoch and N-space for the grasp epoch; why not show results from the same epochs?

B-6) Figure 8 – and the authors' discussion of the driving influence of F5 on M1 activity – how do these findings relate to (Umilta et al., 2007; Spinks et al., 2008)? I think these studies should be explicitly discussed.

Also, is it not surprising to find so little evidence for correspondence between AIP and F5 during planning?

How does the relationship between AIP and F5 that the authors suggest according to their new results compare with prior accounts – e.g. Jeannerod et al., 1995; Fagg and Arbib, 1998?

How do the authors interpret the increased AIP-M1 correspondence during grasp and hold?

B-7) What do the authors mean by the statement regarding M1 on p.26 "…its neuronal population has never been studied explicitly"? What about, for e.g., the work of Umilta et al., 2007; Spinks et al., 2008?

References:

Fagg AH, Arbib MA (1998) Modeling parietal-premotor interactions in primate control of grasping. Neural Netw 11:1277-1303.

Jeannerod M, Arbib MA, Rizzolatti G, Sakata H (1995) Grasping objects: the cortical mechanisms of visuomotor transformation. Trends Neurosci 18:314-320.

Spinks RL, Kraskov A, Brochier T, Umilta MA, Lemon RN (2008) Selectivity for grasp in local field potential and single neuron activity recorded simultaneously from M1 and F5 in the awake macaque monkey. J Neurosci 28:10961-10971.

Umilta MA, Brochier T, Spinks RL, Lemon RN (2007) Simultaneous recording of macaque premotor and primary motor cortex neuronal populations reveals different functional contributions to visuomotor grasp. J Neurophysiol 98:488-501.

---

## [Author Response]

*Essential revisions:*

*The reviewers thought that all comments deserve consideration by the authors. Particularly, the reviewers felt that analyzing the data as a function of the F5 subdivisions would strengthen the study further, particularly since the authors have followed such a strategy in previous work, and we'd hope that such analysis will be straightforward. In addition, the reviewers felt that there were interpretational issues with the role of area AIP that will deserve careful attention.*

*Reviewer #1:*

The study by Schaffelhofer and Scherberger on "Object vision to hand action in macaque parietal, motor, and premotor cortices" reports novel findings on the roles of areas AiP, F5 and M1 in the transformation of visual information to motor action. Simultaneous recordings in these areas from arrays of electrodes were performed. The animals were trained on an excellent grasping task that used a variety of 3D objects and dissociated the different parts of a grasping task into its component parts (e.g. visual, motor planning, motor action). In addition to the use of sophisticated behavioral and recording protocols, the authors also used a glove to quantitatively measure kinematic aspects of grasping movements. The complex data from multiple areas and recording channels as well as movement parameters were analyzed in sophisticated ways to permit novel insight into the transformation of visual signals in areas AIP and F5 to motor planning and action in F5 and M1. I'll leave the detailed critique to the expert reviewer. In focusing on a larger picture in this particular area, I found this paper outstanding in all respects.

Thank you for your extremely favorable evaluation of our paper in particular regarding the larger picture of this research topic.

*Reviewer #2:*

*1) The main limit of this beautiful work is the assumption that area F5 is anatomo-functionally unitary. Recent literature not considered in this work shows that F5 is neither architectonically (Belmalih et al. 2009) nor connectionally (Gerbella et al. 2011) homogeneous: the ventral and anterior part of F5 (called F5a, where the most lateral arrays in the present study have been implanted), hosts 3D-shape-selective and "visual-dominant" neurons (Theys et al. 2012), commonly found also in AIP (Theys et al. 2013) but not in the dorsal part of F5 (i.e. F5p, see Bonini et al. 2014). Furthermore, while in the introduction the authors generally talk about "connections of F5 to the spinal cord and M1", this is true virtually only for F5p (Borra et al. 2010). Considering altogether data recorded from different F5 subdivisions was acceptable in the authors' previous paper (Schaffelhofer et al. 2015, J Neurosci), as its goal was to decode hand configurations, but not here, where the authors want to elucidate the functioning of the cortical grasping network.*

Floating micro-electrode arrays have the advantage of analyzing simultaneously recorded populations on the identical time-line, thereby creating new possibilities of analyzing neural data (e.g. trial-based analysis as shown in Figure 6, Figure 7, Figure 8). However, the technique has the disadvantage of static electrode positions (non-moveable). In other words, a large number of channels are required for acquiring a sufficient sample size of modulated neurons. Primarily for this reason, we have merged the recording sites within area F5 (F5p and F5a). However, we agree of course that F5 consists of specialized sub regions that we have now analyzed and visualized separately (see Figure 3). Specifically, we mapped all identified visual and visuomotor neurons in each of the six electrode arrays. As expected, we found significant differences between the F5 subdivisions consistent with the current literature and your comments. In contrast to F5a, F5p did not (monkey M) or only minimally (monkey Z) contribute to movement planning, but instead was modulated during movement execution.

Beside the new Figure 3, we changed the following parts of the manuscript:

Introduction, second paragraph:

“Connections of the dorsal subdivision of F5 (F5p) to the spinal cord and to M1 provide further evidence of the area’s important role in grasp movement preparation (Borra et al., 2010; Dum and Strick, 2005).“

Motor planning and execution, last paragraph:

“It is notable, that we found significantly different contributions in motor preparation between the F5 recording sites. The visual (Figure 3) and visuomotor modulations (Figure 3) prior to movement primarily originated from the ventral recording array, corresponding to the F5a subdivision (Gerbella et al., 2011; Theys et al., 2012a). In fact, 76% of the visual (abstract objects) and 72% of visuomotor tuned neurons (mixed objects) recorded from F5 were detected on the ventral site (ANOVA p<0.01, all sessions, tested in cue epoch), in line with previously reported enhanced decoding capabilities of planning signals from ventral F5 (Schaffelhofer et al., 2015a). In contrast, the dorsal F5 array, corresponding to F5p, mainly contributed during movement execution by a four-fold increase of its tuned population with respect to the cue epoch.”

Feature coding in area F5, last paragraph:

“As discussed above, premotor preparation activity primarily originated from the ventral F5 array. Thus, the modulations observed prior to movement execution, such as observed in the CDA and hierarchical clustering, mainly based on the ventral recording site. In contrast, both arrays significantly supported movement execution. The different modalities in both recording sites are in line with the architectonical (Belmalih et al., 2009) and connectionally (Borra et al., 2010; Gerbella et al., 2011) distinct subdivisions F5a and F5p, which largely correspond to the ventral and dorsal recording array, respectively. Distinct connections of F5a with SII, AIP and other subdivision of F5 (but not to M1) (Gerbella et al., 2011) suggest an integration of visual, motor and context specific information (Theys et al., 2013). On the other hand, connections of F5p to the hand area of M1 and to the spinal cord (Borra et al., 2010) suggest a rather direct contribution to hand movement control. Taken together, the multimodal preparation signals with visual and motor contribution and the distinct motor feature coding towards planning and execution supports the important role of F5 in visuomotor transformation.”

2) Strictly related to this issue, the electrodes length is unknown. From the authors' previous works (Schaffelhoefer et al. 2015a and Townsend et al. 2011) I infer that "electrode length ranged from 1.5 to 7.1 mm", with the longest electrodes toward the sulcus. Thus, the more superficial electrodes of the lateral arrays certainly sampled F5c motor activity, likely related to mouth and hand movements, (see Graziano et al. 1997 and Maranesi et al. 2012), while the deepest electrodes most likely sampled F5p/a visuomotor activity. Please clarify.

Thank you for reading the manuscript so carefully. We have now added the electrode length in the following sections:

Legend of Figure 1: “…(f) Electrode placement in monkey Z (right hemisphere). Each array consisted of 2 ground and 2 reference electrodes (black), as well as 32 recording channels (white) aligned in a 4x9 matrix. Electrode length for each row increased towards the sulcus from 1.5 – 7.1 mm. (g) Same for monkey M (left hemisphere)…”.

[Please note that Figure 1 has been adapted accordingly to illustrate individual channels]

Chapter “Electrophysiological recordings”:

“In each array, the lengths of the electrodes increased towards the sulcus and ranged from 1.5 (1^st^ row) to 7.1mm (4^th^ row). In area F5, one array was placed in the posterior bank of the inferior arcuate sulcus approximately targeting F5a (longer electrodes)(Theys et al., 2012) and approaching the F5 convexity (F5c; shorter electrodes). The second and more dorsally located array was positioned to target F5p. In AIP, the arrays were implanted into the end of the posterior intraparietal sulcus at the level of area PF and more dorsally at the level of area PFG. In M1, both arrays were placed into the hand area of M1 into the anterior bank of the central sulcus at the level of the spur of the arcuate sulcus (Rathelot and Strick, 2009)”.

We believe that the updated Figure 1–Figure 2 and the corresponding updates in the text now provide full details of the location of the implanted electrodes.

*3) "Data have been analysed from 20 recording sessions (10 per animal)…". I don't understand how many (single and multi) units have been used for the analyses. It seems that a total of about 202 (units)*10 (sessions)=2020 and 355*10=3550 "units" have been considered in the two animals, respectively: with chronic recording this approach would certainly imply some resampling of the same "units" across sessions, increasing their number but reducing the variability of their discharge. At the same time, the authors stated in the Materials and methods section that "all dimensionality reduction results presented for the individual areas originated from the same recording session per monkey", which actually sounds fair, but in contrast with the previous consideration. Please clarify.*

Yes, we recorded in average 202 and 355 neurons from monkey Z and M, respectively (10 sessions per animal, see Results, paragraph 3). The reason we have not stated the summed number of neurons across sessions is exactly because of the resampling of neurons (due to the day-to-day stability of electrodes with respect to their position). The high unit counts you mentioned are correct (591 AIP + 677 M1 + 753 F5 cells = 2021 cells in animal Z, and 892 AIP, 1507 M1 and 1146 F5 cells = 3545 in animal M) but could potentially give the reader the incorrect impression about the sum of *unique*neurons and we therefore avoided them. Thus, to provide the fairest unit counts, we stated the average cell count per session and averaged analysis results accordingly (see Figure. 3, Figure 7, Figure 8).

For results that reflect individual sessions, we have added the number of included cells to each individual Figure (please see Figure 2, Figure 4, Figure 5, and 6 and their corresponding supplements). For the single session analysis, we have explicitly mentioned “all dimensionality reduction results are from the same recording session” to make clear that we did not select the “best” representations we could find within 10 sessions for each of the individual areas (AIP, F5 and M1), but instead presented populations from the same recording session. This step should increase the comparability of results between individual areas.

Moreover, we have made clear that single and multi-units have been analyzed together. See chapter “Electrophysiological recordings”, last sentence: “No other pre-selection was applied and single and multiunit activity have been analysed together.”

The manuscript now provides detailed information for each analysis with respect to unit count.

Somewhat related to your comment, we also solved a question that has been important to us. We added a variation of the hierarchical cluster analysis (HCA) in order to express the coding of task conditions across recording sessions (in addition to individual sessions). For this, we measured the Mahalanobis distance between each pair of group means (e.g., between condition 00 and 11) resulting in a 50 x 50 distance matrix. These matrices, observed from each individual session, were averaged across all recording sessions and then used as a basis for an across-session HCA. The resulting dendrograms are now shown as supplemental figures in addition to the results of individual example sessions (Figure 2—figure supplement 1–Figure 2—figure supplement 2; Figure 5—figure supplement 1–Figure 5—figure supplement 2, Figure 6—figure supplement 1–Figure 6—figure supplement 2). The following sections of the manuscript were altered:

Chapter “Hierarchical Cluster Analysis”, end of chapter:

“Additionally, we averaged the Mahalanobis distance matrices of all individual session to build dendrograms expressing the N-space of an animal across all recording sessions (e.g. Figure 2—figure supplement 1 for AIP of animal Z).”

Chapter “Vision for hand action”, end of paragraph 3:

“Importantly, consistent results were observed in both monkeys when performing HCA across all recording sessions (Figure 2—figure supplement 1–Figure 2—figure supplement 2, see Materials and methods).”

Chapter “Feature coding in area F5”, end of paragraph 7:

“HCA performed on the averaged population response across all recording sessions (see Materials and methods) confirmed in both animals the motor characteristics of simultaneously recorded populations in single sessions (Figure 5—figure supplement 1–Figure 5—figure supplement 2).

Chapter “Feature coding in area M1”, last paragraph:

“The similarity of J- and N-space of M1 was not only visible in the first two components, but was also quantified across all dimensions in the hierarchical cluster trees of the simultaneously recorded M1 population (Figure 6) and when averaging the population response across all recording sessions (Figure 6—figure supplement 1-2 for monkey M and Z, resp.).”

*4) I recommend using the same y-axis scales for plots in Figure 3). Nevertheless, it is clear that the tuning profile of AIP does not change when abstract or mixed sets of objects are used (see Figure 3): I totally understand the authors' reasoning, but they will acknowledge that all the abstract objects share an elongated shape, while those of the mixed set are much more variable. Thus, the lack of difference in the AIP tuning profile between the two sets is strange and should be better explained/discussed. For example, could it be due to receptive fields properties of AIP neurons (see Romero et al. 2014; Romero and Janssen 2016)?*

Thank you for pointing this out. We updated the y-axis scales according to your suggestion.

The similar quantitative responses (in terms of percentage of significantly tuned neurons) for the abstract and mixed shapes, in our opinion, are not strange but strikingly extend our findings from the complete object set. We understand that a higher variation in object shape could have affected the number of tuned cells, in stark contrast to the observed minor differences presented in Figure 3. However, monkeys viewed the objects from the side, which substantially reduced the apparent visual uniformity of the abstract object set (less elongated profiles). Furthermore, we report in Figure 3 the percentage of significantly tuned neurons, and not, for example, the average tuning strength (e.g., the difference between the strongest and weakest response in each object set), which might have been a more sensitive measure for detecting absolute tuning differences between both object sets.

*Reviewer #3:*

*Altogether, I consider the work exceptional. The data make new and important contributions to understanding the cortical mechanisms underlying grasp control. By concurrently recording and characterising the relationships between limb kinematics and neural activity from AIP, F5, and M1 while monkeys viewed and grasped a wide range of objects, the scope of the current investigation is unparalleled in the current primate neurophysiological literature. Also outstanding are the beautiful data visualisation methods that the authors advance here.*

*I have a few suggestions, however, detailed below. My hope is that the authors will find these comments constructive.*

*A) Main concerns:*

*(A-1) The role of AIP*

*I'm having some difficulty understanding the authors’ stance on the role of AIP. I believe they are arguing that their data suggest that AIP is encoding object properties in visual terms, and not in motoric terms. They suggest that the results of their CDA show that AIP encodes objects according to shape in visual space, independent of the way they are grasped.*

*My interpretation of their stance is that AIP activity does not reflect action possibilities in sensorimotor terms – but rather reflects the 'flagging' of visual features relevant for actions, but NOT (yet) represented as spatiomotor plans.*

*If this is indeed their stance, I see this as a significant departure from prior conceptualizations (Jeannerod et al., 1995).*

However, I remain confused about this argument, both (1) conceptually, and (2) with respect to how some aspects of their data can be viewed as consistent these claims.

Thank you for your detailed thoughts that we found very constructive. Following your comments, we have clarified the interpretation/summary of AIP throughout the manuscript and highlighted the corresponding sections in green (see section “Vision for action”, paragraph 3 and 6, “Discussion”, paragraph 5-7). We hope that these revisions improve the general understanding of the AIP results and their interpretation. Here we first summarize our stance and then address point-by point your questions regarding concept and data:

Grasping an object in different ways does not only require different motor plans, but also differentiated visual descriptions of the object. For example, when grasping a coffee cup, we need to create a geometrical description of the cup and an additional visual selection of its object parts (e.g., the handle) that we intend to grasp (e.g., perform a hook grip). Our data suggests that AIP is a dynamical visual area that might be responsible for differentiating and selecting these objects parts in a visual (geometrical) rather than in a motor space.

*1) Conceptually. The authors talk about "object affordances" as different from "motor affordances".*

*To me, affordances = action possibilities; they are sensory representations of the external environment that are inseparably linked to the motor apparatus and capabilities of the perceiving animal. In other words, affordances are, according to my understanding, inextricably motor.*

*What does it mean to "extract object affordances relevant for grasping", but to not at the same time encode those same objects according to the spatiomotor properties required for their manual interaction?*

*They state: "…suggest encoding of object affordances rather than motor affordances".*

I urge the authors to better clarify what is meant by object vs. motor affordances.

Your comment is directly related to comment (d) of Reviewer #2. With object affordances we intended to express the visual extraction of object features of an object part relevant for grasping. To avoid confusion, we replaced “object affordances” with “object features” throughout the manuscript.

*2) Data. Some aspects of AIP data suggest encoding related to motoric features.*

*a) Shift for same object – big horizontal cylinders – when grasped from below vs. above.*

b) Significant differentiation of conditions 00 vs. 01.

c) Motor similarity measures – Figure 7 – presumably a measure of.75 for AIP is different than a measure of "1" – and thus, does suggest some correspondence between N- and J-space encoding in AIP

*d) Also, there interpretation of Figure 3 vs. d – "…observed encoding of abstract objects in AIP can be explained exclusively by object shape."*

*Is it not possible that these data may reflect the fact that abstract versus mixed object set have more graspable parts – a greater number of possible actions –, and AIP encoding is sensitive to these differences?*

*This account could be considered consistent with the authors’ interpretations regarding the evidence for reduced tuning in AIP over time for abstract shapes: "…reduction of 3D shapes to the relevant parts for grasping…".*

*Summary:*

*While the authors seem to argue for a strict separation in roles between AIP and F5 – visual vs. motor, respectively – I see the data as more likely evidence for a continuum – a graded difference – visual prevalence in AIP, with motor encoding vs. motoric prevalence in F5, with visual encoding.*

*I suggest that the authors clarify their stance, and/or temper their arguments.*

Describing the neural differentiation in cases a-b as motor features would be seductive and in agreement with current literature. However, in our opinion, this standpoint does not reflect the global view on our data. For the following reasons we interpret the observed modulations in AIP as a visual rather than a motoric differentiation:

In both animals, the AIP population separated objects primarily on visual object attributes at the session- (Figure 2) and multi-session level (Figure 2—figure supplement 1–Figure 2—figure supplement 2): objects clustered primarily based on visual attributes, such as shape and size, although they were grasped in highly different fashions. In contrast, motor attributes were rarely found.Differences with respect to the selected grip type were only apparent when *the same* object was grasped differently (e.g., handle box, horizontal cylinder), whereas all other objects maintained their shape clusters. In contrast, motor feature coding, as observed in F5 and M1, would separate all objects in motor terms (not only those that offer different object parts). This could indicate visual rather than motor processes for grasping.The separation of such conditions (handle, cylinder) originated from the same shape cluster (see Figure 2 and Figure 9) also suggesting visual rather than motor transformations.

**Author response image 1. fig9:** Visualization of (**a**) reduced and (**b**) complete population show perfectly overlapping clusters of small (black triangles, 51-54) and large cylinders (red triangles, 55-56) in the early cue phase (100 ms after cue onset). These conditions start separating over time as shown in Figure 2 and Figure 2. **DOI:**
http://dx.doi.org/10.7554/eLife.15278.022

In d), the reviewer suggested that abstract objects might have more graspable parts, and hence a larger number of possible actions, than objects of the mixed set. However, we regard this as rather unlikely. Neither the grasp variability of both animals nor the visual aspect of the objects justifies the conclusion that abstract objects have more graspable parts. In fact, we specifically designed the abstract set such that objects have many different visual parts, but only one graspable part, which both animals also uniformly selected.

However, we agree with Reviewer #3 that the visual and motor space cannot be separated completely, as we have discussed previously (Schaffelhofer et al. 2015). For example, coming from the motor side, Figure 6 shows the separation of objects based on the recorded hand kinematics. Although this is a pure motor representation of objects, some shapes tend to cluster together, such as the vertical cylinders or the cubes, due to the design of the objects that shape the hand. In other words, even a pure motor representation creates similarities with the visual space and vice versa.

Taken together, and in agreement with reviewer #3, the differences between AIP and F5 can only be described as graded ones, with visual prevalence in AIP and motoric prevalence in F5. While we maintain our general interpretation of the AIP results, we tempered our arguments, as suggested. Both, visual and motor separation of objects is now discussed in the Results and Discussion section (we highlighted all text edits in the manuscript with respect to this comment in green).

“Vision for hand action”, paragraph 3:

“Hierarchical cluster analysis (HCA) performed on these distance measures confirmed the findings of the CDA and revealed a clear clustering according to object shape during visual presentation of the object (Figure 2) that widely remained during movement execution, although with significantly shorter neural distances (Figure 2).”

“Vision for hand action”, paragraph 7:

“This reduced selectivity could indicate either motor or visual transformations that are both required for grasping: First, the abstract objects were grasped with the same hand configuration (see Figure 3). […] Together, these observations suggest a visual rather than motor representation in AIP.”

“Discussion – Visual processing for grasping”, paragraph 3 and 4:

“We emphasize that visual coding at the population level does not preclude motor coding of some individual cells, as suggested previously (Murata et al., 1997). Rather, the evidence of bidirectional connections to F5 suggests that subpopulations in AIP exist that reflect feedback motor signals from F5.”

“These modulations could reflect either motor (Fagg and Arbib, 1998) or visual transformations required for grasping. […] These differentiation schemes support our hypothesis of visual rather than motor transformations in AIP.”

*B) Intermediate concerns*

B-1) in the Introduction, the authors state that prior work did not dissociate between visual and motor features. At first, I thought this was inconsistent with the work of Sakata and colleagues – describing visual dominant, visuomotor, and motor dominant cell types. I think some discussion of this work upfront, including how the current approach critically differs, would be helpful.

Previous approaches separated neurons into visual, visuomotor and motor cells primarily based on their responses to objects in light or dark condition (activation or no activation). According to Sakata and co-workers, Motor dominant cells become activated during grasping and holding in both light and dark. They do not fire during object fixation. Visual-motor neurons discharge stronger during grasping in light than in the dark. Visual-dominant neurons discharge during object fixation and when grasping in light. Thus, the epoch of activation has been the main criteria for classification.

Our work took a significantly different approach by comparing the coding (modulation patterns) of neurons and of neuronal populations with respect to highly variable objects and the measured hand kinematics in order to separate visual from motor processes. Our work demonstrated that activations in visual epochs can reflect motor processing, whereas grasp modulation can reflect object geometries.

We followed your suggestion and entirely revised paragraph 4-6 of the Introduction:

“To create a deeper understanding of how visual information is transformed into motor commands, a precise identification and differentiation of visual and motor processes within the grasping network is required. […] Our data revealed distinct roles of the grasping network in translating visual object attributes (AIP) into planning (F5) and execution signals (M1) and allowed visualizing the propagation of these features for grasping. “

*B-2) Introduction – last paragraph – it isn't clear to me what the authors mean to say here, without first reading the rest of the paper. I think this section could be made clearer.*

We revised the last paragraph accordingly (see above). We believe, that the concept and intention of our experiments are now better explained.

B-3) I'm not sure I understand their explanation of the shape-specificity prior to fixation. Please try to make this clearer.

Shape-specificity during the fixation epoch is likely due to our block design, in which we presented objects of similar shape on the same turntable. This fact is a small detail, however it could raise concerns on whether coding in AIP was really due to object shape, or rather to the object presentation order (different turntables presented sequentially). We revised our arguments to make this clearer:

“The large number of objects presented in one recording session required separating the 48 objects on different turntables (see Figure 1), often objects of similar shape. This separation created small offsets already in the fixation epoch, but at very low modulations. An extreme case is shown in Figure 2. This might raise concerns on whether coding in AIP was really due to object shape, or rather to the object presentation order (different turntables presented sequentially). However, shape-wise clustering in AIP cannot be explained by the task design for the following reasons: (1) The offsets in the fixation epoch were very small in comparison to the visual modulations observed in AIP when the objects were presented (e.g., see Video 5). (2) The set of “mixed” objects – presented and grasped in the same block – clustered with other objects of the same shape (e.g. ring in mixed block clusters with other ring objects). Together, this demonstrates clear shape processing in AIP.”

B-4) At first I thought I missed something – it seemed as though AIP was not looked at in terms of J- and N-space encoding in the same way that F5 and M1 were. However, later it is clear that they did analyse AIP in the same way. Why not also show these data, as they do for F5 (Figure 4, Figure 5) and M1 (Figure 6)?

For the complete N-space the AIP, F5, and M1 populations were analyzed and visualized in the same way. In the reduced space the areas were visualized differently to represent the full dimensional space (hierarchical clustering) best. That is, showing the object separation of AIP in 3D to reveal the object-shape clusters best, whereas F5 and M1 populations were shown in 2D and aligned (rotated) to the J-space (with PCRA) to allow direct comparison of both representations next to each other. We believe that this kind of visualization reflects and explains the full-dimensional results best.

B-5) Figure 4 – it is unclear to me why they plot J-space for the hold epoch and N-space for the grasp epoch; why not show results from the same epochs?

Premotor area F5 provided higher modulations (separation of conditions) in the grasping epoch, when the hand was approaching the object. This is also shown and discussed in Video 5, where you can see F5 expressions earlier than in M1. We therefore selected this epoch in area F5 to represent motor execution. However, all epochs, including the hold epoch, are shown below in Figure 4.

B-6) Figure 8 – and the authors' discussion of the driving influence of F5 on M1 activity – how do these findings relate to (Umilta et al., 2007; Spinks et al., 2008)? I think these studies should be explicitly discussed.

We find both studies most helpful. Umilta et al. and Spinks et al. report similarity between F5 and AIP with respect to grasp coding at the LFP and single unit level in support of our feature code correlation results. We now discuss and reference their work in the following sections:

Chapter “Feature coding in area M1”, first paragraph:

“In agreement with the general population response (Figure 3), single neurons of M1 were almost exclusively modulated during movement execution and showed minimal modulations during preparatory epochs (Umilita et al. 2007), as also indicated by the example neuron in Figure 6.”

“Discussion”, last paragraph:

“The high similarity of the F5 and M1 population during movement execution (Figure 4 vs. 6e, Figure 8, and Video 5) is in agreement with findings of similar coding schemes between both areas in spiking (Umilta et al., 2007) and β-band LFP activity (Spinks et al., 2008).”

*Also, is it not surprising to find so little evidence for correspondence between AIP and F5 during planning?*

The applied analysis reflects the coding similarity between both areas, not their physical connectivity. In other words, when two areas share the same feature coding with respect to the 50 conditions the correlation coefficient increases. In fact, the correlation analysis results, although exploring the data with an entirely different method, match the identified visual and motor coding (and their proportion) from the population analysis. In detail, only an F5 subpopulation coded objects in visual terms, and therefore similar with respect to AIP (see Figure 3). This explains the relatively low feature-communication peaks between both areas, and demonstrates that the coding of objects is represented strongly differently in AIP and F5 during large parts of the task, although both areas are highly modulated by these objects. Please note that we would expect significantly higher feature communication between AIP and F5 when precisely targeting the F5a subdivision, which holds predominantly visual-dominant neurons, in comparison to F5p (Theys et al. 2012), and directly receives information from AIP (Gerbella et al. 2010).

In response to your comment, we added the following sentences to chapter “Results/Feature code correlation”:

“These results are consistent with the proportion of F5 visual cells identified in animal Z and M (Figure 3), which only temporarily shared the visual coding with AIP. Interestingly, F5 and AIP demonstrated minimal similarities during movement planning, which could support the different encoding of visual and motor features described on the population level (Figure 2, Figure 4,Figure 5).”

How does the relationship between AIP and F5 that the authors suggest according to their new results compare with prior accounts – e.g. Jeannerod et al., 1995; Fagg and Arbib, 1998?

Existing models of the grasping network share the same basic interpretation of the F5-AIP network as the main circuit for visuomotor transformation (Jeannerod et al., 1995, Fagg and Arbib, 1998, Rizzolati et Luppino, 2001). In support, our data provides important electrophysiological evidence that object features are indeed transformed into motor representations between both areas.

Previous models were, however, uncertain on how and where visual information is exactly translated into motor commands and whether both areas perform both visual and motor computations. Fagg et Arbib (1998) and Rizzolatti et Luppino (2001) consider grasp transforms already in AIP. They propose that AIP is using the visual input to generate multiple affordances passed to F5, which selects the final grip type.

Our data suggest a slightly different information flow in which visual and motor processes are organized more separately. In this understanding, AIP is responsible for providing object properties to F5 (F5a) thereby activating motor solutions that, in return to AIP, help further extracting geometrical features relevant for grasping. F5 with its connection to visual input (F5a) and motor output (F5p), could be the main hub for coordinating visual feature extraction with AIP and motor planning and subsequent execution with M1.

We believe that our edits in response to this reviewer’s comment a) (see above) addresses this question.

*How do the authors interpret the increased AIP-M1 correspondence during grasp and hold?*

This comment is related to comment (1b)-d. As discussed above, visual representations of objects and motor features of the hand cannot be entirely disassociated, because the object geometries define the shape of the grasping hand. Even pure motor representations, such as the recorded hand kinematics, can therefore share similarities with an object space. When M1 changes from a mostly unmodulated (planning) to a highly modulated state (grasp epoch) the similarity to an object space increases. In other words, we interpret the increase of similarity as a result of the highly increased modulations in M1.

*B-7) What do the authors mean by the statement regarding M1 on p.26 "…its neuronal population has never been studied explicitly"? What about, for e.g., the work of Umilta et al., 2007; Spinks et al., 2008?*

We tempered this statement: “Although the motor relevance of the hand area of M1 has been described extensively with electrophysiological (Schieber, 1991; Schieber and Poliakov, 1998; Spinks et al., 2008; Umilta et al., 2007) and anatomical methods (Dum and Strick, 2005; Rathelot and Strick, 2009), it has been unclear how versatile hand configurations are encoded at the population level.”